# Development, validation, and application of a machine learning model to estimate salt consumption in 54 countries

**Wilmer Cristobal Guzman-Vilca[1,2,3], Manuel Castillo-Cara[4], Rodrigo M Carrillo-Larco[2,5]***

[1]School of Medicine Alberto Hurtado, Universidad Peruana Cayetano Heredia, Lima, Peru; [2]CRONICAS Centre of Excellence in Chronic Diseases, Universidad Peruana Cayetano Heredia, Lima, Peru; [3]Sociedad Científica de Estudiantes de Medicina Cayetano Heredia (SOCEMCH), Universidad Peruana Cayetano Heredia, Lima, Peru; [4]Universidad de Lima, Lima, Peru; [5]Department of Epidemiology and Biostatistics, School of Public Health, Imperial College London, London, United Kingdom

**\*For correspondence:**
r.carrillo-larco@imperial.ac.uk

**Competing interest:** The authors declare that no competing interests exist.

**Abstract** Global targets to reduce salt intake have been proposed, but their monitoring is challenged by the lack of population-based data on salt consumption. We developed a machine learning (ML) model to predict salt consumption at the population level based on simple predictors and applied this model to national surveys in 54 countries. We used 21 surveys with spot urine samples for the ML model derivation and validation; we developed a supervised ML regression model based on sex, age, weight, height, and systolic and diastolic blood pressure. We applied the ML model to 54 new surveys to quantify the mean salt consumption in the population. The pooled dataset in which we developed the ML model included 49,776 people. Overall, there were no substantial differences between the observed and ML-predicted mean salt intake (p<0.001). The pooled dataset where we applied the ML model included 166,677 people; the predicted mean salt consumption ranged from 6.8 g/day (95% CI: 6.8–6.8 g/day) in Eritrea to 10.0 g/day (95% CI: 9.9–10.0 g/day) in American Samoa. The countries with the highest predicted mean salt intake were in the Western Pacific. The lowest predicted intake was found in Africa. The country-specific predicted mean salt intake was within reasonable difference from the best available evidence. An ML model based on readily available predictors estimated daily salt consumption with good accuracy. This model could be used to predict mean salt consumption in the general population where urine samples are not available.

## Editor's evaluation

Salt intake is a major determinant of volume status, blood pressure values, and congestion, but its estimation is challenging because of the need of measuring 24-h urinary sodium excretion over a number of days, which is unfeasible in most countries. The demonstration of the feasibility of estimating accurately salt intake at the population level using artificial intelligence starting from simple and widely available variable is therefore important for epidemiological and intervention studies in which salt intake is a major player, particularly, but not only, in countries experiencing economic hardships.

## Introduction

The association between high sodium/salt intake and high blood pressure, a major risk factor of cardiovascular diseases (CVDs), is well-established (*He et al., 2013*; *World Health Organization, 2021a*; *Poggio et al., 2015*). More than 1.7 million CVD deaths were attributed to a diet high in sodium in 2019, with ~90% of these deaths occurring in low- and middle-income countries (LMICs) (*GBD 2019 Risk Factors Collaborators, 2020*; *GBD Results Tool, 2021*). Consequently, salt reduction has been included in international goals: the World Health Organization (WHO) recommendation of limiting salt consumption to <5 g/day (*World Health Organization, 2021a*), and the agreement by the WHO state members of a 30% relative reduction in mean population salt intake by 2025 (*WHO. World Health Organization, 2021*). Because available evidence suggests that sodium/salt consumption is higher than the global targets (*Powles et al., 2013*; *Carrillo-Larco and Bernabe-Ortiz, 2020*; *Oyebode et al., 2016*) we need timely and consistent data of sodium/salt consumption in the general population to track progress of salt reduction targets.

Global efforts have been made to produce comparable estimates of sodium/salt intake for all countries (*Powles et al., 2013*). Similarly, researchers have summarized all the available evidence in specific world regions (*Carrillo-Larco and Bernabe-Ortiz, 2020*, *Oyebode et al., 2016*). Although the global endeavor was based on the gold standard method to assess sodium/salt intake (i.e., 24 hr urine sample), their estimates were up to 2010 (*Powles et al., 2013*). Therefore, robust and comparable sodium/salt intake estimates for all countries lack for the last 10 years. The regional endeavors summarized population-based evidence, yet they conducted study-level meta-analyses in which the original studies could have followed different laboratory methods, and they did not study all countries in the region. Therefore, comparability across studies could be limited and evidence lacks for many countries. Finding a method to estimate sodium/salt consumption in national samples leveraging on available data is needed to update and complement the existing evidence (*Powles et al., 2013*; *Carrillo-Larco and Bernabe-Ortiz, 2020*; *Oyebode et al., 2016*; *Thout et al., 2019*). Quantifying sodium/salt intake based on 24 hr urine samples is costly and burdensome, limiting its use in population-based studies or national health surveys. As an alternative, equations have been developed to estimate sodium/salt intake based on spot urine (SU) samples (*Brown et al., 2013*; *Kawasaki et al., 1993*; *Toft et al., 2014*; *Tanaka et al., 2002*). Although these equations may not deliver identical results to those based on 24 hr urine samples at the individual level, at the population level the difference between SU samples and 24 hr samples appears to be small (*Huang et al., 2016*; *Santos et al., 2020*). However, these equations have been used in few WHO STEPS and other national health surveys (*World Health Organization, 2021b*), leaving several countries without data to quantify the local sodium/salt consumption because they do not have access to SU samples (*World Health Organization, 2021c*).

If we could (accurately) estimate sodium/salt intake at the population level based on variables that are routinely available in national health surveys (e.g., weight or blood pressure), mean sodium/salt intake at the population level in countries that currently lack urine data (i.e., 24 hr or spot) could be computed using these available predictors. Advanced analytic techniques like machine learning (ML) could make accurate predictions and inform about the mean sodium/salt intake at the population level. We developed an ML predictive model to estimate mean salt intake at the population level (not at the individual level) using routinely available variables in national health surveys. We applied this ML model to other national health surveys without urine data to compute the mean salt intake in the general population.

## Results

### Study population for model derivation and validation

The pooled dataset included 49,776 people from 21 surveys in 19 countries (i.e., two countries, Bhutan and Mongolia, had two surveys) conducted between 2013 and 2019 (*Appendix 1—table 1*). Overall, the mean age ranged from 33 (95% confidence interval [95% CI]: 33–34) years in Zambia to 43 (95% CI: 42–44) years in Belarus. The proportion of men ranged from 35.7% in Tonga to 61.4% in Solomon Islands. The mean SBP was lowest in Jordan (117.7 mmHg [95% CI: 115.7–119.8 mmHg]) and highest in Belarus (134.6 mmHg [95% CI: 133.6–135.5 mmHg]). The mean DBP was lowest in Chile (73.6 mmHg [95% CI: 72.5–74.6 mmHg]) and highest in Belarus (84.9 mmHg [95% CI: 84.4–85.5 mmHg]). The

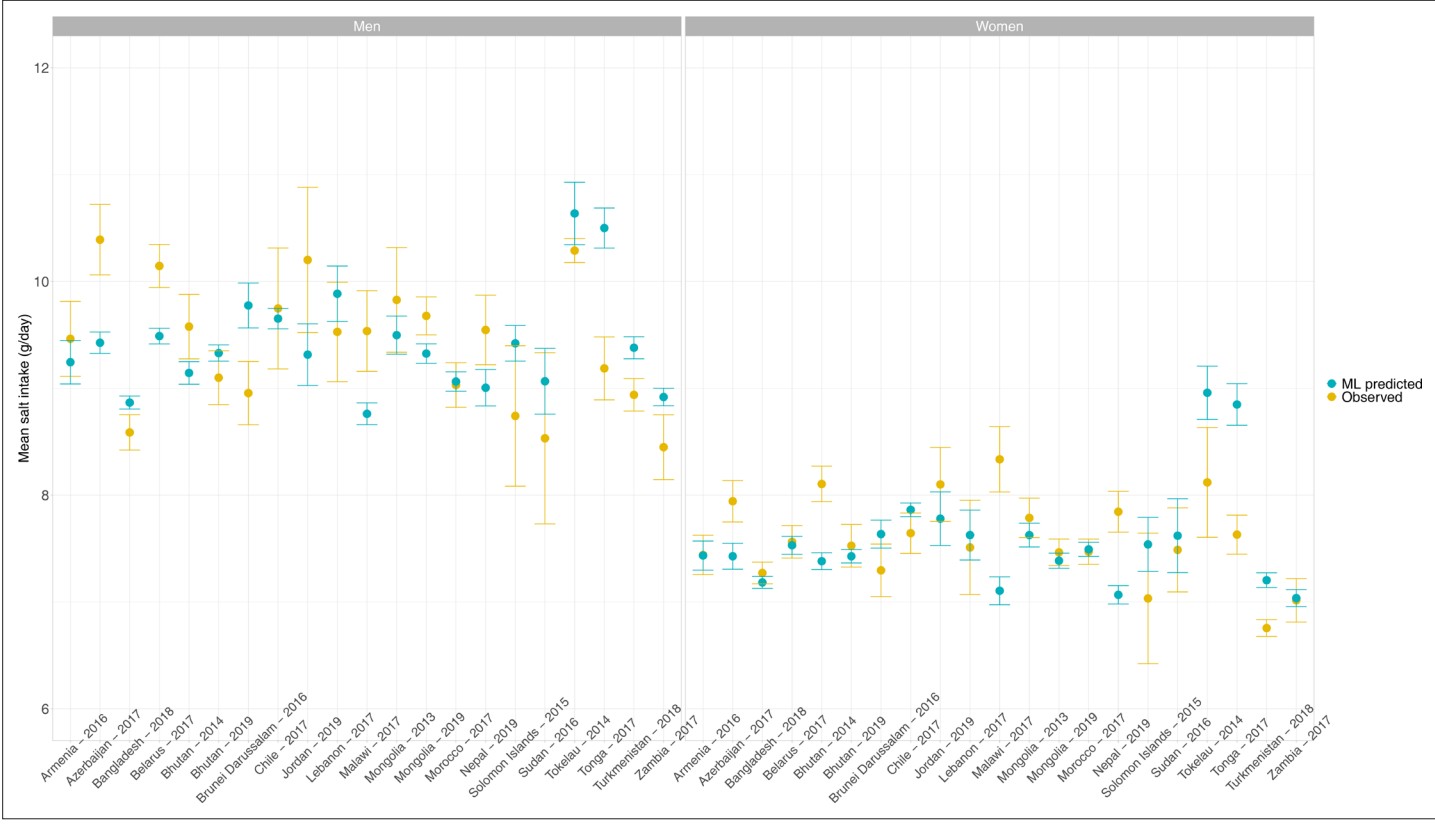

**Figure 1.** Observed and predicted mean salt intake (g/day) by sex in each survey included in the machine learning (ML) model development. Exact estimates (along with their 95% CI) are presented in *Appendix 1—table 2*. These results were computed with the test dataset only. Results are for the HuR algorithm, which was the model with the best performance.

mean weight ranged from 54.6 kg (95% CI: 53.8–55.5 kg) in Nepal to 98.6 kg (95% CI: 97.7–99.5 kg) in Tonga. The mean height ranged from 1.55 m (95% CI: 1.55–1.56 m) in Nepal to 1.71 m (95% CI: 1.70–1.71 m) in Tokelau.

## Observed and predicted mean salt intake during the ML model derivation and validation

In the test dataset including 20 WHO STEPS surveys and one national health survey (Chile) (i.e., 21 surveys in total), the observed mean salt intake computed as per the INTERSALT equation was higher in men than in women in all countries; it ranged from 8.5 g/day (95% CI: 8.2–8.8 g/day; Zambia) to 10.4 g/day (95% CI: 10.1–10.7 g/day; Azerbaijan) in men and from 6.8 g/day (95% CI: 6.7–6.8 g/day; Turkmenistan) to 8.3 g/day (95% CI: 8.0–8.6 g/day; Malawi) in women. Across countries, the predicted mean salt intake was also higher in men than in women. Results for each survey are presented in *Figure 1* and *Appendix 1—table 2*.

The mean observed salt intake was higher in people aged ≥30 years (7.9 g/day vs. 8.4 g/day, p<0.05 for independent *t*-test), and so was for people with raised blood pressure (≥140/90 mmHg) (8.7 g/day vs. 8.2 g/day, p<0.05). The mean salt consumption was also different across body mass index (BMI) categories (p<0.05 for ANOVA test). The same profile was found for predicted mean salt intake (*Appendix 1—table 3*).

In men across all countries in the test dataset including 20 WHO STEPS surveys (representing 18 countries) and 1 national health survey (Chile), the mean difference between observed and predicted mean salt intake was –0.02 g/day (p<0.001 for paired *t*-test). Across all surveys, the positive mean difference farthest from zero was 0.54 g/day (Nepal, p<0.001 for paired *t*-test), and the negative mean difference farthest from zero was –1.31 g/day (Tonga, p<0.001 for paired *t*-test). The mean difference closest to zero was –0.03 g/day (Morocco, p=0.308 for paired *t*-test) (*Appendix 1—table 4*).

In women across all countries in the test dataset including 20 WHO STEPS surveys (representing 18 countries) and 1 national health survey (Chile), the mean difference between the observed and predicted mean salt intake was 0.01 g/day (p<0.001 for paired *t*-test). The positive mean difference farthest from zero was 1.23 g/day (Malawi, p<0.001 for paired *t*-test) and the negative mean difference farthest from zero was in –1.22 g/day (Tonga, p<0.001 for paired *t*-test). The mean difference closest to zero was 0.01 g/day (Armenia, p=0.195 for paired *t*-test) (*Appendix 1—table 4*).

None of the countries herein analyzed, regardless of the method of sodium intake assessment (i.e., observed or predicted), showed a mean salt intake below the WHO recommended level of <5 g/day (*Figure 1*, *Appendix 1—table 2*). The same occurred for the mean salt intake estimates using the Kawasaki, Toft, and Tanaka formulas (*Appendix 1—table 5*).

## Implementation of the developed ML model to predict salt consumption in 54 countries

The pooled dataset where we applied the ML model included 166,677 people from 54 countries in 54 WHO STEPS surveys conducted between 2004 and 2018 (*Appendix 1—table 6*). Overall, the mean age ranged from 31 (95% CI: 31–32) years in Ethiopia to 43 (95% CI: 40–47) years in Barbados. The proportion of men ranged from 17.2% in Eritrea to 63.8% in Timor-Leste. The mean SBP was lowest in Cambodia (116.2 mmHg [95% CI: 115.6–116.9 mmHg]) and highest in Mozambique (138.7 mmHg [95% CI: 136.3–141.0 mmHg]). The mean DBP was lowest in Cambodia (72.4 mmHg [95% CI:

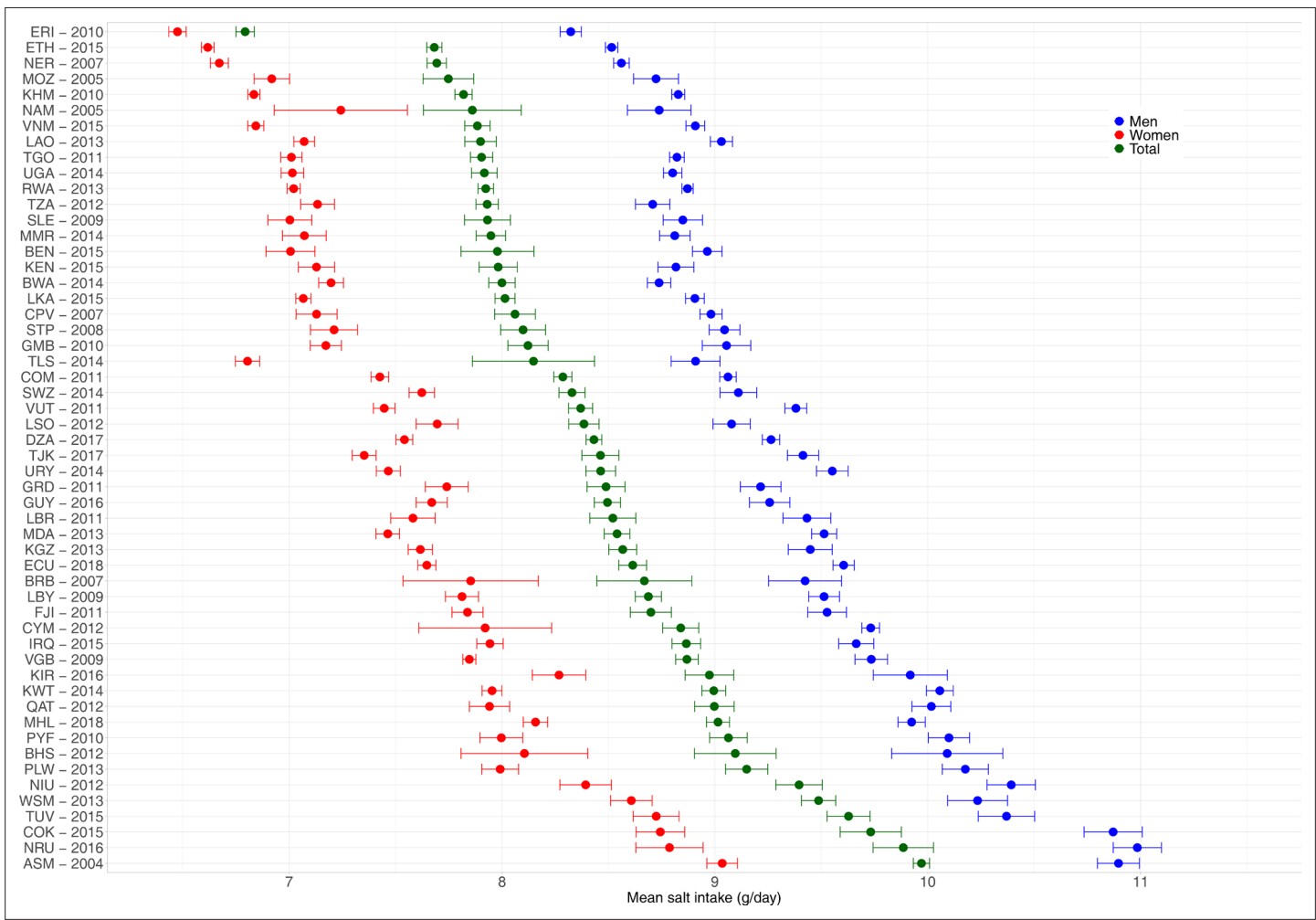

**Figure 2.** Predicted mean salt intake (g/day) by sex in each of the 54 national surveys included in the application of the model herein developed. Exact estimates (along with their 95% CI) are presented in *Appendix 1—table 7*. Countries are presented in ascending order based on their overall mean salt intake (i.e., countries with the highest mean salt intake are at the bottom).

71.8–73.0 mmHg]) and highest in Kyrgyzstan (86.8 mmHg [95% CI: 85.9–87.8 mmHg]). The mean weight ranged from 51.8 kg (95% CI: 51.2–52.4 kg) in Eritrea to 100.4 kg (95% CI: 100.1–100.8 kg) in American Samoa. The mean height ranged from 1.54 m (95% CI: 1.54–1.55 m) in Lao People's Democratic Republic to 1.70 m (95% CI: 1.70–1.71 m) in British Virgin Islands.

Across the 54 countries, the overall predicted mean salt intake ranged from 6.8 g/day (95% CI: 6.8–6.8 g/day) in Eritrea to 10.0 g/day (95% CI: 9.9–10.0 g/day) in American Samoa. The mean was always higher in men than in women. None of the countries herein analyzed, regardless of sex, showed a predicted mean salt intake below the WHO recommended level of <5 g/day (*Figure 2*, *Appendix 1—table 7*).

In men, the countries with the highest predicted mean salt intakes were Nauru (11.0 g/day), American Samoa and Cook Islands (both with 10.9 g/day), and Niue and Tuvalu (both with 10.4 g/day); remarkably, all of these countries are in the Western Pacific. In contrast, the lowest predicted mean salt intake in men was in Eritrea (8.3 g/day), Ethiopia (8.5 g/day), and Niger (8.6 g/day); remarkably, all of these countries are in Africa.

In women, the countries with the highest predicted mean salt intake were American Samoa (9.0 g/day), Nauru (8.8 g/day), and Cook Islands and Tuvalu (both with 8.7 g/day); all of these countries are in the Western Pacific. Conversely, the lowest predicted mean salt intake in women was in Eritrea (6.5 g/day), Ethiopia (6.6 g/day), and Niger (6.7 g/day); all of these countries are in Africa.

## Discussion
### Main findings
This work leveraged on 21 national health surveys and readily available predictors to develop an ML model to predict salt consumption; this model was then applied to national surveys in 54 countries. It should be noted that we analyzed SU samples. These are not the gold standard to assess salt consumption. Results should be interpreted in light of this limitation, considering that our model aimed to deliver estimates at the population level (not individual level) (*Huang et al., 2016*; *Santos et al., 2020*). The HuR ML algorithm yielded the predictions closest to the observed salt intake: the mean difference between predicted and observed salt consumption across surveys was –0.02 g/day in men and 0.01 g/day in women. We used this novel ML model to predict salt consumption in 54 countries, where the mean salt consumption ranged from 8.3 g/day (Eritrea) to 11.0 g/day (Nauru) in men; these numbers in women ranged from 6.5 g/day (Eritrea) to 9.0 g/day (American Samoa). This work aimed to elaborate on novel analytical tools to predict salt consumption where national surveys have not collected this information, limiting their ability to keep track of mean sodium consumption in the general population. Pending external independent validation, our model could be used in monitoring frameworks of salt consumption because most countries do not collect sodium samples in their national health surveys. Our model could contribute to the global surveillance of salt consumption, a relevant cardiometabolic risk factor (*He et al., 2013*; *World Health Organization, 2021a*; *Poggio et al., 2015*).

### Public health implications
ML models have been used extensively to predict relevant clinical outcomes (e.g., mortality) and epidemiological indicators (e.g., forecasting COVID-19 cases) (*Wang et al., 2020*; *Wynants et al., 2020*; *Groot et al., 2021*; *Watson et al., 2021*; *Mohan et al., 2021*). Furthermore, ML algorithms have proven to be useful for understanding complex outcomes (e.g., identifying clusters of people with diabetes) based on simple predictors (e.g., BMI) in nationally representative survey data (*Oh et al., 2019*; *García de la Garza et al., 2021*; *Carrillo-Larco et al., 2021*). Our work complements the current evidence on ML algorithms by demonstrating its use in a relevant field: population salt consumption. In so doing, we delivered a pragmatic tool that could be used to inform the surveillance of salt consumption in countries where national surveys do not objectively collect this information (e.g., SU samples). Moreover, this work provided preliminary evidence to update the global estimates of population-based sodium consumption (*Powles et al., 2013*) by informing about the mean sodium consumption in 54 countries. Our results suggest that mean salt consumption is above the WHO recommended level in all the 54 countries herein analyzed, and it was the highest among countries in the Western Pacific, and the lowest among countries in Africa. This finding, which is consistent with

a global work (*Powles et al., 2013*), calls for urgent actions to reduce salt consumption in these 54 countries, especially those in the Western Pacific.

We do not believe that our – or any other – ML model should replace a comprehensive population-based nationally representative health survey with 24 hr or SU samples. However, until such surveys are available in many countries and periodically conducted, we could suggest using an estimation approach to shed lights about the mean salt consumption in the population. Our ML model seems to be a reasonably good alternative and could become a pragmatic tool for surveillance systems that keep track of sodium consumption in accordance with global goals (*World Health Organization, 2021a*; *WHO. World Health Organization, 2021*).

## Research in context

A global effort provided mean sodium/salt consumption estimates for 187 countries in 1990 and 2010 (*Powles et al., 2013*); they used 24 hr urine samples and dietary reports from surveys conducted in 66 countries. Unfortunately, their results were until 2010. Our results advanced this evidence by providing more recent salt consumption estimates because most of the surveys in which we applied our ML model were conducted after 2010 (*Appendix 1—table 6*).

Compared to the global estimates for the same countries in 2010, (*Powles et al., 2013*), our mean salt consumption estimates were very similar. For example, our 2010 mean salt consumption estimates for Cambodia, Eritrea, and the Gambia were 7.8 g/day, 6.8 g/day, and 8.1 g/day, whereas the estimates by *Powles et al., 2013* were 11.0 g/day, 5.9 g/day, and 7.7 g/day (*Appendix 1—table 8*; *Powles et al., 2013*). We further compared our estimates for surveys conducted between 2007 and 2013 (±3 years around 2010) with the 2010 estimates provided by *Powles et al., 2013*, and our results were also within reasonable difference. The largest differences were in Tajikistan (8.5 by our ML model vs. 13.5 by *Powles et al., 2013*), as well as in Kyrgyzstan (8.6 vs. 13.4 by *Powles et al., 2013*) and Samoa (9.5 vs. 5.2 by *Powles et al., 2013*). It appears that our predictions were higher than those provided by *Powles et al., 2013* in countries with presumably low salt consumption (e.g., Samoa); conversely, in countries with presumably high salt consumption (e.g., Kyrgyzstan), our predictions revealed smaller estimates than those by *Powles et al., 2013* (*Appendix 1—table 8*). These differences could be explained by the fact that our ML model was developed based on SU samples rather than 24 hr urine samples as *Powles et al., 2013* did. Strong evidence indicates that estimates based on SU may overestimate salt intake at lower levels of consumption and underestimate salt intake at higher levels of consumption (*Huang et al., 2016*).

In addition to the global work by *Powles et al., 2013*, there are other reports from some specific countries. For example, a survey conducted between 2012 and 2016 with 24 hr urine samples in Fiji and Samoa showed that the mean salt consumption was 10.6 g/day and 7.1 g/day, respectively (*Santos et al., 2019*). The estimates from our ML model for Fiji (2011) and Samoa (2013) suggested that the mean salt consumption was 8.7 g/day and 9.5 g/day, respectively. A survey in Vanuatu in 2016 based on 24 hr urine sample informed that the mean salt intake was 5.9 g/day (*Paterson et al., 2019*); our estimate for the year 2011 was 8.4 g/day. In 2009 in Vietnam, a survey with SU samples revealed that the mean salt consumption was 9.9 g/day (*Jensen et al., 2018*); our prediction for the year 2015 was 7.9. These comparisons suggest that our ML-predicted estimates are plausible and close to the best available evidence.

Although these comparisons do not validate our predictions in the 54 national surveys, they suggest that our salt consumption estimates are within reasonable distance from the best available evidence. Until better data are available (e.g., national survey with spot or 24 hr urine sample), our model could provide preliminary evidence to inform the national mean salt consumption. Careful interpretation is warranted to understand the strengths and limitations of our ML-based predictions.

## Strengths and limitations

We followed sound and transparent methods to develop an ML model to predict salt consumption at the individual level. We leveraged on open-access national data collected following standard and consistent protocols (*World Health Organization, 2021b*; *Departamento de Epidemiologia. Ministerio de Salud, 2021*). Most of the surveys we analyzed were conducted after 2010, providing more recent evidence than the latest global effort to quantify salt consumption in all countries (*Powles et al., 2013*). Notwithstanding, we must acknowledge some limitations. First and foremost, urine data

was based on a spot sample, which is not the gold standard (24 hr urine sample) to measure daily salt consumption. Future work should verify and advance our results using on 24 hr urine samples available in nationally representative samples; in the meantime, our work has led the foundations and hopefully sparked interest to use available data and novel analytical techniques to deliver estimates of salt consumption in the general population. While SU samples may not be the best approach to estimate salt consumption at the individual level, at the population level the means estimated based on SU samples and 24 hr urine samples are similar (*Huang et al., 2016*; *Santos et al., 2020*). Therefore, the limitation of using SU samples only may have had little impact on our mean estimates, which are the country level, not at the individual level. While this – reanalysis of SU sample rather than 24 hr urine samples – is a limitation of our work, it is also an observation showing the lack of nationally representative surveys with 24 hr urine samples available for independent reanalyses. Second, even though we analyzed 21 national surveys (representing 19 countries) to develop our ML model, the sample size could still be limited for a data-driven ML algorithm (i.e., 24,889 observations were included in model development). A larger and global work in which all relevant data sources are pooled is needed; while this endeavor takes place, our work has provided recent estimates of salt consumption at the population level in 54 countries. In this line, there are still countries that were not herein included. Researchers in these countries, along with local (e.g., ministries of health) and international health authorities (e.g., WHO), should conduct studies/surveys to collect data on salt consumption. This would inform global targets but also local needs and interventions.

An ML model based on readily available variables was accurate to predict daily salt consumption. This ML model applied to 54 national surveys with no urine samples to compute daily salt consumption revealed high levels of salt intake particularly in the Western Pacific region. Pending further validation, this ML model could be used to keep track of the overall sodium consumption where resources are not available to conduct national surveys with urine samples.

## Methods

### Study design

This is an individual-level data pooling ML analysis.

### Data sources

We sought surveys that met these two criteria: (i) nationally representative health surveys (i.e., community or subnational surveys were not included); and (ii) surveys that were open access or that could be accessed without significant administrative burden (e.g., data sharing agreements that may involve institutional signatures).

First, we downloaded 20 WHO STEPS surveys and 1 national health survey with SU samples; these surveys were used for the training, validation, and testing of the ML model. These 21 surveys represented 19 countries; two countries contributed with two surveys: Bhutan 2014 and 2019 as well as Mongolia 2013 and 2019. Second, we downloaded 54 new WHO STEPS surveys that had the variables included in the ML prediction model (see 'Variables' section), but did not have SU samples. The ML model herein developed was applied to these 54 surveys to estimate the mean salt consumption in the population.

To identify additional data sources, we searched the original publications included in one global analysis (*Powles et al., 2013*) and three systematic reviews about sodium/salt consumption at the population level (*Carrillo-Larco and Bernabe-Ortiz, 2020*; *Oyebode et al., 2016*; *Thout et al., 2019*). This search led to the identification of the national health survey included in the model derivation. All other data sources included in those references (*Powles et al., 2013*; *Carrillo-Larco and Bernabe-Ortiz, 2020*; *Oyebode et al., 2016*; *Thout et al., 2019*) did not meet our selection criteria.

In conclusion, our ML model was developed based on 21 surveys (20 WHO STEPS and 1 national health survey). Then, our ML model was applied to 54 WHO STEPS survey to compute the mean daily salt consumption at the population level.

According to the World Bank classification (*Appendix 1—table 9*), there were 9 high-income countries (2 in model derivation and 7 in model application), 16 low-income countries (1 in model derivation and 15 in model application), 26 lower-middle-income countries (9 in model derivation and 17 in model application), and 18 upper-middle-income countries (6 in model derivation and 12

model application). There were four countries (one in model derivation and three in model application) without income classification (British Virgin Islands, Cook Islands, Niue, and Tokelau).

## Rationale

We hypothesized that an ML model could accurately predict salt consumption at the individual level, to then inform the overall mean in the underlying population. In addition, we endeavored to develop an ML model with simple predictors; that is, variables that are routinely available in national health surveys contrary to urine sample that are seldom collected in national health surveys. If the model were indeed accurate, then it could be applied to national surveys without urine samples but with the relevant predictors to inform about the mean salt consumption in the overall population. These model-driven estimates could be preliminary until a national health survey is conducted to study mean salt consumption with urine samples. Ideally, salt consumption should be informed by 24 hr urine samples, which are seldom available in large population-based and nationally representative health surveys. The fact that we analyzed SU samples is a limitation of our work, and the results should be interpreted accordingly. However, we aimed to develop an ML model that can be used to predict mean estimates at the population level, not at the individual level. In other words, our model should not be applied to a patient to estimate his/her salt consumption. We did not develop a diagnostic tool to replace SU or 24 hr urine samples. Our model should be applied to survey data to compute the mean sodium/salt consumption in the population (not in individuals). Empirical evidence suggests that, at the population level, mean estimates based on SU samples and on 24 hr urine samples are similar (*Huang et al., 2016*; *Santos et al., 2020*).

## Variables

The predictors we used in the ML model were sex, age (years), weight (kg), height (m), systolic blood pressure (SBP, mmHg), and diastolic blood pressure (DBP, mmHg).

The analyzed surveys collect anthropometric and three blood pressure measurements. These are taken by trained fieldworkers following a standard protocol (*World Health Organization, 2021b*; *Departamento de Epidemiologia. Ministerio de Salud, 2021*). We used measured weight and height to compute the BMI ($kg/m^2$). We used the mean SBP and mean DBP of the second and third blood pressure measurements (i.e., the first blood pressure measurement was discarded).

The outcome was salt intake as per the INTERSALT equation (*Brown et al., 2013*). We chose this equation because it has been used by WHO STEPS surveys. There is a specific INTERSALT equation for each sex, and they both include the following variables: age (years), BMI ($kg/m^2$), SU sodium (mmol/L), and SU creatinine (mmol/L) (*Brown et al., 2013*). We used the following sex-specific formulas:

$$Men: \left\{ 23.51 + \left[ 0.45 \text{ x } Na_{SU} \right] - \left[ 3.09 \text{ x } Cr_{SU} \right] + \left[ 4.16 \text{ x BMI} \right] + \left[ 0.22 \text{ x age} \right] \right\}$$

$$Women: \left\{ 3.74 + \left[ 0.33 \text{ x } Na_{SU} \right] - \left[ 2.44 \text{ x } Cr_{SU} \right] + \left[ 2.42 \text{ x BMI} \right] + \left[ 2.34 \text{ x age} \right] - \left[ 0.03 \text{ x age}^2 \right] \right\}$$

where the subscript *SU* indicates spot urine, *Na* is sodium, *Cr* is creatinine, and *BMI* is body mass index. Because some STEPS surveys had SU creatinine in mg/dL, these values were multiplied by 0.00884 to obtain SU creatinine in mmol/L. No conversion was needed for sodium in SU samples because all surveys herein included already had urinary sodium in mmol/L. The INTERSALT equation computes 24 hr sodium intake, which is then divided by 17.1 to obtain the salt intake in grams per day (g/d) (*Brown et al., 2013*). For descriptive purposes, we also computed salt intake based on the *Kawasaki et al., 1993*, *Toft et al., 2014*, and *Tanaka et al., 2002* equations. Of note, our outcome variable was informed by SU samples and not by 24 hr urine samples (gold standard to assess salt consumption). Results should be interpreted according to this limitation.

## Analysis

### Data preparation

Our complete-case analysis was restricted to men and nonpregnant women aged between 15 and 69 years because of data availability. We dropped participants with implausible BMI levels (outside the range 10–80 $kg/m^2$) or with implausible weight (outside the range 12–300 kg) or height records (outside the range 1.00–2.50 m). Participants with SBP outside the range 70–270 mmHg were discarded, and so were participants with DBP outside the range 30–150 mmHg. We excluded

records with SU creatinine <1.8 or > 32.7 mmol/L for males and <1.8 or >28.3 for females (*Santos et al., 2019*; *Paterson et al., 2019*). In addition, we excluded participants with estimated salt intake (using the four equations) above or below 3 standard deviations from the equation-specific mean (*Appendix 1—figure 1*; *Jensen et al., 2018*). After completing data preparation, observations were randomly assigned from the pooled dataset (100%) into three datasets for the ML analysis: training dataset (50%), test dataset (30%), and validation dataset (20%).

## Machine learning modeling

Our research aim was a regression problem where we had a known outcome attribute (salt consumption at the subject level). Therefore, we planned a supervised ML regression analysis. Details about the modeling process are available in the 'Extended methods' (Appendix 2). In brief, we designed a work pipeline with five steps. First, *data analysis*, where we dropped missing observations, we explored the available data to choose scaling and transformation methods to secure all variables were in the same scale or units, and we also planned transformations for categorical variables (e.g., one-hot encoding). Second, *feature importance analysis*, where we investigated the contribution of each predictor to the regression model through methods like Random Forest (RF) and Recursive Feature Elimination. The aim of this second step was to exclude any predictor that would not contribute to the regression model. Notably, all predictors (see 'Variables' section) chosen following expert knowledge were kept in the analysis (i.e., the feature importance analysis did not suggest the exclusion of any predictor). Third, *data processing*, having explored the available data (first step in the work pipeline), we implemented different scaling and transformation methods (e.g., Box-Cox, principal component analysis and polynomial features). Fourth, *data modeling*, where we implemented 10 ML algorithms: (i) linear regression (LiR); (ii) Hubber regressor (HuR); (iii) ridge regressor (RiR); (iv) multilayer perceptron (MLP); (v) support vector regressor (SVR); (vi) $k$-nearest neighbors (KNN); (vii) RF; (viii) gradient boost machine (GBM); (ix) extreme gradient boosting (XBG); and (x) a customized neural network. All these ML algorithms performed similarly, so the decision to choose one was postponed to the fifth (last) step in the work pipeline. Up to this point, we used the training and validations datasets. Five, *forecasting* of the predicted attribute in new data (i.e., data not used for model training); in this step, we used the test dataset to choose the model that yielded predictions closest to the observed salt intake. Results comparing the observed and the predicted salt intake were computed in the test dataset alone. For each country, we ran a paired $t$-test between the observed and predicted salt consumption, where a difference was deemed significant at a $p<0.05$. We also computed the absolute difference between the observed and predicted salt intake. We chose the HuR algorithm because it showed the mean difference closest to zero in both sexes combined (observed – predicted = 0) (*Appendix 2—table 2*, *Appendix 2—figure 3*) . All summary estimates (e.g., mean salt intake) were computed accounting for the complex survey design of the surveys included in the analysis.

## Application of the developed ML model

Having developed the ML model following the steps above described, we applied the model to 54 WHO STEPS national surveys that did not have urine samples but included the predictors in the ML model (see 'Variables' section). In each of these 54 surveys, we computed the mean daily salt intake accounting for the complex survey design. These surveys were preprocessed following the same procedures described in the 'Data preparation' section.

## Role of the funding source

The funder had no role in the study design, analysis, interpretation, or decision to publish. The authors are collectively responsible for the accuracy of the data. The arguments and opinions in this work are those of the authors alone, and do not represent the position of the institutions to which they belong.

## Additional information

### Funding

| Funder | Grant reference number | Author |
|---|---|---|
| Wellcome Trust | 214185/Z/18/Z | Rodrigo M Carrillo-Larco |

The funders had no role in study design, data collection and interpretation, or the decision to submit the work for publication.

### Author contributions

Wilmer Cristobal Guzman-Vilca, Conceptualization, Data curation, Formal analysis, Investigation, Methodology, Software, Visualization, Writing – original draft, Writing – review and editing; Manuel Castillo-Cara, Conceptualization, Data curation, Formal analysis, Investigation, Methodology, Software, Validation, Visualization, Writing – original draft, Writing – review and editing; Rodrigo M Carrillo-Larco, Conceptualization, Data curation, Funding acquisition, Investigation, Methodology, Supervision, Writing – original draft, Writing – review and editing

### Author ORCIDs

Wilmer Cristobal Guzman-Vilca http://orcid.org/0000-0002-2194-8496
Manuel Castillo-Cara http://orcid.org/0000-0002-2990-7090
Rodrigo M Carrillo-Larco http://orcid.org/0000-0002-2090-1856

### Ethics

We did not seek approval by an Institutional Review Board. We used individual-level survey data that do not include any personal identifiers.

### Decision letter and Author response

Decision letter https://doi.org/10.7554/eLife.72930.sa1
Author response https://doi.org/10.7554/eLife.72930.sa2

## Additional files

### Supplementary files
• Transparent reporting form
• Source code 1. Analysis Code | Python and R.

### Data availability

This study used nationally-representative survey data that are in the public domain, which was requested through the online repository (https://extranet.who.int/ncdsmicrodata/index.php/home). We provide the analysis code of data preparation and data analysis as supplementary materials to this paper (Source Code File - "Analysis Code | Python and R").

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

# Appendix 1

**Appendix 1—table 1.** Weighted distribution of predictors in each survey included in the machine learning model development.

| Country | Year | Sample size | Mean age (years) | Age range (years) | Proportion of men (%) | Mean, minimum and maximum values of SBP (mmHg) | Mean, minimum, and maximum values of DBP (mmHg) | Mean, minimum, and maximum values of weight (kg) | Mean, minimum, and maximum values of height (m) | Mean, minimum, and maximum values of urinary sodium (mmol/L) | Mean, minimum, and maximum values of urinary creatinine (mmol/L) |
|---|---|---|---|---|---|---|---|---|---|---|---|
| Armenia | 2016 | 1074 | 40 | 18–69 | 49.7 | 129 (86–238) | 85 (49–148) | 70.9 (35–139) | 1.66 (1.27–1.89) | 128.6 (10.6–237.6) | 10.1 (1.9–27.3) |
| Azerbaijan | 2017 | 2359 | 39 | 18–69 | 49.5 | 126 (82–230) | 81 (48–142) | 73.1 (36–174) | 1.67 (1.15–1.98) | 167.7 (2–389) | 11.9 (1.8–31.8) |
| Bangladesh | 2018 | 6200 | 39 | 18–69 | 46.9 | 121 (72–251) | 79 (32–147) | 55.9 (28–111) | 1.56 (1–2.11) | 119.6 (4–422) | 8.4 (2.2–32.3) |
| Belarus | 2017 | 4503 | 43 | 18–69 | 47.1 | 135 (88–257) | 85 (54–147) | 77.7 (41–144) | 1.7 (1.05–1.99) | 149.5 (10.5–371.4) | 12.2 (1.8–32.7) |
| Bhutan | 2014 | 6163 | 38 | 18–69 | 59.2 | 126 (75–228) | 85 (46–142) | 61.4 (23–115) | 1.6 (1.11–1.96) | 142.1 (6–388) | 8.1 (1.9–29.7) |
| Bhutan | 2019 | 6163 | 34 | 15–69 | 56.8 | 124 (85–224) | 82 (44–137) | 61.9 (28.5–140) | 1.58 (1.07–1.92) | 129.9 (4.7–444.9) | 10.4 (1.8–32.7) |
| Brunei Darussalam | 2016 | 1635 | 35 | 18–69 | 51.4 | 123 (76–218) | 78 (46–138) | 69 (31.2–138.3) | 1.59 (1.32–1.84) | 122.6 (19.9–329) | 12.6 (1.8–32.6) |
| Chile | 2017 | 2952 | 39 | 15–69 | 49.8 | 120 (81–226) | 74 (44–130) | 75.7 (38.3–146.9) | 1.63 (1.34–1.96) | 135.8 (10–324) | 12.1 (1.8–32.2) |
| Jordan | 2019 | 1040 | 37 | 18–69 | 50.2 | 118 (75–200) | 78 (50–120) | 76.3 (35.5–159.5) | 1.66 (1.36–1.95) | 165.4 (13–365) | 13.6 (1.8–32.5) |
| Lebanon | 2017 | 998 | 42 | 17–69 | 48.7 | 129 (80–214) | 77 (35–123) | 78.3 (40–141) | 1.68 (1.2–1.96) | 124.4 (4–385) | 11.5 (1.9–32) |
| Malawi | 2017 | 1601 | 35 | 18–69 | 56.4 | 122 (74–222) | 76 (40–142) | 58.5 (33.6–119) | 1.61 (1.36–1.96) | 186.5 (11–399.9) | 10.7 (1.9–32.4) |
| Mongolia | 2013 | 7505 | 42 | 15–64 | 50.3 | 129 (88–220) | 82 (50–134) | 71 (30.6–138) | 1.62 (1.27–1.92) | 134.1 (13.1–515) | 10.9 (1.8–31.9) |
| Mongolia | 2019 | 7505 | 36 | 15–69 | 50.9 | 120 (76–254) | 77 (48–143) | 68.4 (29–159) | 1.64 (1.34–1.98) | 117 (2.1–348.9) | 7.5 (1.8–28.3) |
| Morocco | 2017 | 3435 | 40 | 18–69 | 50.6 | 128 (83–228) | 78 (40–139) | 70.9 (35–168) | 1.66 (1.34–1.95) | 122.3 (26.3–575.2) | 10.3 (1.8–31.4) |
| Nepal | 2019 | 2560 | 36 | 15–69 | 41 | 124 (81–239) | 81 (55–146) | 54.6 (26–160) | 1.55 (1.21–2.03) | 140.9 (3–437) | 5.6 (1.8–25.5) |
| Solomon Islands | 2015 | 172 | 38 | 18–69 | 61.4 | 121 (88–188) | 77 (52–104) | 67.9 (38.5–122) | 1.61 (1.41–1.8) | 99.3 (7–250) | 9.7 (1.9–28.4) |
| Sudan | 2016 | 571 | 36 | 18–69 | 55.9 | 128 (89–231) | 85 (58–132) | 72.2 (35.6–174) | 1.67 (1.42–1.92) | 128.5 (5–459) | 14 (1.9–32.4) |
| Tokelau | 2014 | 181 | 35 | 18–63 | 56 | 125 (76–184) | 79 (53–128) | 94.8 (58–158.3) | 1.71 (1.16–1.88) | 62.4 (20–265) | 5 (2–7.7) |
| Tonga | 2017 | 755 | 40 | 18–69 | 35.7 | 131 (96–208) | 83 (53–148) | 98.6 (48.1–181) | 1.69 (1.4–1.94) | 101.9 (4–327) | 15.3 (1.8–32.7) |
| Turkmenistan | 2018 | 3584 | 37 | 18–69 | 52.7 | 127 (88–268) | 83 (54–149) | 72.4 (39–142) | 1.68 (1.16–1.98) | 109.2 (10–163) | 11.1 (4.5–18.3) |
| Zambia | 2017 | 2488 | 33 | 18–69 | 50.3 | 125 (73–248) | 77 (36–148) | 60.9 (33.8–150) | 1.62 (1.01–2.07) | 137.2 (10–375) | 12.2 (1.8–32.4) |

**Appendix 1—table 2.** Observed and predicted mean salt intake (g/day) by sex in each survey included in the machine learning model development.

| Country | Year | Sex | Mean salt intake | Mean salt intake lower 95% confidence interval | Mean salt intake upper 95% confidence interval | Category |
|---|---|---|---|---|---|---|
| Armenia | 2016 | Men | 9.24 | 9.04 | 9.45 | ML predicted |
| Armenia | 2016 | Men | 9.46 | 9.11 | 9.81 | Observed |

*Appendix 1—table 2 Continued on next page*

*Appendix 1—table 2 Continued*

| Country | Year | Sex | Mean salt intake | Mean salt intake lower 95% confidence interval | Mean salt intake upper 95% confidence interval | Category |
|---|---|---|---|---|---|---|
| Armenia | 2016 | Women | 7.43 | 7.3 | 7.57 | ML predicted |
| Armenia | 2016 | Women | 7.44 | 7.26 | 7.62 | Observed |
| Azerbaijan | 2017 | Men | 9.43 | 9.33 | 9.53 | ML predicted |
| Azerbaijan | 2017 | Men | 10.39 | 10.06 | 10.72 | Observed |
| Azerbaijan | 2017 | Women | 7.43 | 7.31 | 7.55 | ML predicted |
| Azerbaijan | 2017 | Women | 7.94 | 7.75 | 8.14 | Observed |
| Bangladesh | 2018 | Men | 8.87 | 8.8 | 8.93 | ML predicted |
| Bangladesh | 2018 | Men | 8.59 | 8.42 | 8.75 | Observed |
| Bangladesh | 2018 | Women | 7.18 | 7.13 | 7.24 | ML predicted |
| Bangladesh | 2018 | Women | 7.27 | 7.17 | 7.37 | Observed |
| Belarus | 2017 | Men | 9.49 | 9.42 | 9.56 | ML predicted |
| Belarus | 2017 | Men | 10.14 | 9.94 | 10.35 | Observed |
| Belarus | 2017 | Women | 7.53 | 7.45 | 7.61 | ML predicted |
| Belarus | 2017 | Women | 7.56 | 7.41 | 7.72 | Observed |
| Bhutan | 2014 | Men | 9.14 | 9.04 | 9.25 | ML predicted |
| Bhutan | 2014 | Men | 9.58 | 9.27 | 9.88 | Observed |
| Bhutan | 2014 | Women | 7.38 | 7.3 | 7.46 | ML predicted |
| Bhutan | 2014 | Women | 8.1 | 7.94 | 8.27 | Observed |
| Bhutan | 2019 | Men | 9.33 | 9.25 | 9.41 | ML predicted |
| Bhutan | 2019 | Men | 9.1 | 8.85 | 9.35 | Observed |
| Bhutan | 2019 | Women | 7.43 | 7.36 | 7.49 | ML predicted |
| Bhutan | 2019 | Women | 7.53 | 7.33 | 7.73 | Observed |
| Brunei Darussalam | 2016 | Men | 9.78 | 9.57 | 9.99 | ML predicted |
| Brunei Darussalam | 2016 | Men | 8.95 | 8.66 | 9.25 | Observed |
| Brunei Darussalam | 2016 | Women | 7.64 | 7.5 | 7.77 | ML predicted |
| Brunei Darussalam | 2016 | Women | 7.3 | 7.05 | 7.54 | Observed |
| Chile | 2017 | Men | 9.65 | 9.56 | 9.75 | ML predicted |
| Chile | 2017 | Men | 9.75 | 9.18 | 10.31 | Observed |
| Chile | 2017 | Women | 7.86 | 7.8 | 7.93 | ML predicted |
| Chile | 2017 | Women | 7.64 | 7.45 | 7.83 | Observed |
| Jordan | 2019 | Men | 9.31 | 9.03 | 9.6 | ML predicted |
| Jordan | 2019 | Men | 10.2 | 9.52 | 10.88 | Observed |
| Jordan | 2019 | Women | 7.78 | 7.53 | 8.03 | ML predicted |
| Jordan | 2019 | Women | 8.1 | 7.75 | 8.45 | Observed |
| Lebanon | 2017 | Men | 9.88 | 9.62 | 10.14 | ML predicted |
| Lebanon | 2017 | Men | 9.53 | 9.06 | 9.99 | Observed |

*Appendix 1—table 2 Continued on next page*

*Appendix 1—table 2 Continued*

| Country | Year | Sex | Mean salt intake | Mean salt intake lower 95% confidence interval | Mean salt intake upper 95% confidence interval | Category |
|---|---|---|---|---|---|---|
| Lebanon | 2017 | Women | 7.63 | 7.39 | 7.86 | ML predicted |
| Lebanon | 2017 | Women | 7.51 | 7.07 | 7.95 | Observed |
| Malawi | 2017 | Men | 8.76 | 8.66 | 8.86 | ML predicted |
| Malawi | 2017 | Men | 9.54 | 9.16 | 9.91 | Observed |
| Malawi | 2017 | Women | 7.1 | 6.97 | 7.24 | ML predicted |
| Malawi | 2017 | Women | 8.34 | 8.03 | 8.64 | Observed |
| Mongolia | 2013 | Men | 9.5 | 9.32 | 9.68 | ML predicted |
| Mongolia | 2013 | Men | 9.83 | 9.34 | 10.32 | Observed |
| Mongolia | 2013 | Women | 7.63 | 7.51 | 7.74 | ML predicted |
| Mongolia | 2013 | Women | 7.79 | 7.6 | 7.97 | Observed |
| Mongolia | 2019 | Men | 9.32 | 9.23 | 9.42 | ML predicted |
| Mongolia | 2019 | Men | 9.68 | 9.5 | 9.85 | Observed |
| Mongolia | 2019 | Women | 7.39 | 7.32 | 7.46 | ML predicted |
| Mongolia | 2019 | Women | 7.46 | 7.34 | 7.59 | Observed |
| Morocco | 2017 | Men | 9.06 | 8.97 | 9.15 | ML predicted |
| Morocco | 2017 | Men | 9.03 | 8.82 | 9.24 | Observed |
| Morocco | 2017 | Women | 7.49 | 7.43 | 7.56 | ML predicted |
| Morocco | 2017 | Women | 7.47 | 7.35 | 7.59 | Observed |
| Nepal | 2019 | Men | 9 | 8.83 | 9.18 | ML predicted |
| Nepal | 2019 | Men | 9.55 | 9.22 | 9.87 | Observed |
| Nepal | 2019 | Women | 7.07 | 6.98 | 7.15 | ML predicted |
| Nepal | 2019 | Women | 7.84 | 7.65 | 8.04 | Observed |
| Solomon Islands | 2015 | Men | 9.42 | 9.25 | 9.59 | ML predicted |
| Solomon Islands | 2015 | Men | 8.74 | 8.08 | 9.4 | Observed |
| Solomon Islands | 2015 | Women | 7.54 | 7.29 | 7.79 | ML predicted |
| Solomon Islands | 2015 | Women | 7.03 | 6.42 | 7.64 | Observed |
| Sudan | 2016 | Men | 9.07 | 8.76 | 9.37 | ML predicted |
| Sudan | 2016 | Men | 8.53 | 7.73 | 9.33 | Observed |
| Sudan | 2016 | Women | 7.62 | 7.27 | 7.97 | ML predicted |
| Sudan | 2016 | Women | 7.49 | 7.09 | 7.88 | Observed |
| Tokelau | 2014 | Men | 10.64 | 10.34 | 10.93 | ML predicted |
| Tokelau | 2014 | Men | 10.29 | 10.18 | 10.4 | Observed |
| Tokelau | 2014 | Women | 8.96 | 8.71 | 9.21 | ML predicted |
| Tokelau | 2014 | Women | 8.12 | 7.61 | 8.63 | Observed |
| Tonga | 2017 | Men | 10.5 | 10.31 | 10.69 | ML predicted |
| Tonga | 2017 | Men | 9.19 | 8.89 | 9.48 | Observed |

*Appendix 1—table 2 Continued on next page*

*Appendix 1—table 2 Continued*

| Country | Year | Sex | Mean salt intake | Mean salt intake lower 95% confidence interval | Mean salt intake upper 95% confidence interval | Category |
|---------|------|-----|------------------|------------------------------------------------|------------------------------------------------|----------|
| Tonga | 2017 | Women | 8.85 | 8.65 | 9.04 | ML predicted |
| Tonga | 2017 | Women | 7.63 | 7.45 | 7.81 | Observed |
| Turkmenistan | 2018 | Men | 9.38 | 9.28 | 9.48 | ML predicted |
| Turkmenistan | 2018 | Men | 8.94 | 8.79 | 9.09 | Observed |
| Turkmenistan | 2018 | Women | 7.2 | 7.13 | 7.27 | ML predicted |
| Turkmenistan | 2018 | Women | 6.76 | 6.68 | 6.83 | Observed |
| Zambia | 2017 | Men | 8.92 | 8.84 | 9 | ML predicted |
| Zambia | 2017 | Men | 8.45 | 8.15 | 8.75 | Observed |
| Zambia | 2017 | Women | 7.04 | 6.96 | 7.12 | ML predicted |
| Zambia | 2017 | Women | 7.01 | 6.81 | 7.22 | Observed |

ML: machine learning; SBP: systolic blood pressure; DBP: diastolic blood pressure.

**Appendix 1—table 3.** Observed and predicted mean salt intake (g/day) by age, body mass index (BMI) category, and blood pressure status across all surveys included in the machine learning model development dataset.

| Attributed | Salt consumption (g/day) observed using surveys included in the derivation model | | Salt consumption (g/day) estimated using the surveys included in the derivation model | |
|------------|------|------|------|------|
| | Mean | p-Value for independent *t*-test or ANOVA test | Mean | p-Value for independent *t*-test or ANOVA test |
| Age <30 years | 7.9 | | 8.0 | |
| Age ≥ 30 years | 8.4 | <0.001 | 8.3 | < 0.001 |
| BMI <18.5 kg/m$^2$ | 7.0 | | 7.0 | |
| BMI 18.5–24.9 kg/m$^2$ | 7.8 | | 7.7 | |
| BMI 25.0–29.9 kg/m$^2$ | 8.6 | | 8.4 | |
| BMI ≥ 30 kg/m$^2$ | 9.3 | < 0.001 | 9.3 | < 0.001 |
| Raised blood pressure ( ≥ 140/90 mmHg) | 8.7 | | 8.6 | |
| No raised blood pressure | 8.2 | < 0.001 | 8.1 | < 0.001 |

These results do not consider the survey sampling design.

**Appendix 1—table 4.** Mean difference (g/day) between observed and predicted salt intake by sex in each survey included in the machine learning (ML) model development.

| Country | Year | Sex | Mean difference | Mean difference lower 95% confidence interval | Mean difference upper 95% confidence interval | p-Value |
|---------|------|-----|-----------------|-----------------------------------------------|-----------------------------------------------|---------|
| Armenia | 2016 | Men | 0.22 | –0.06 | 0.5 | 0.0007 |
| Armenia | 2016 | Women | 0.01 | –0.12 | 0.13 | 0.1953 |
| Azerbaijan | 2017 | Men | 0.96 | 0.67 | 1.26 | < 0.0001 |
| Azerbaijan | 2017 | Women | 0.52 | 0.37 | 0.66 | < 0.0001 |

*Appendix 1—table 4 Continued on next page*

*Appendix 1—table 4 Continued*

| Country | Year | Sex | Mean difference | Mean difference lower 95% confidence interval | Mean difference upper 95% confidence interval | p-Value |
|---|---|---|---|---|---|---|
| Bangladesh | 2018 | Men | –0.28 | –0.44 | –0.12 | < 0.0001 |
| Bangladesh | 2018 | Women | 0.09 | –0.01 | 0.19 | 0.0004 |
| Belarus | 2017 | Men | 0.66 | 0.47 | 0.84 | < 0.0001 |
| Belarus | 2017 | Women | 0.03 | –0.09 | 0.16 | 0.6258 |
| Bhutan | 2014 | Men | 0.43 | 0.17 | 0.7 | < 0.0001 |
| Bhutan | 2014 | Women | 0.72 | 0.57 | 0.88 | < 0.0001 |
| Bhutan | 2019 | Men | –0.23 | –0.48 | 0.02 | 0.0007 |
| Bhutan | 2019 | Women | 0.1 | –0.08 | 0.28 | 0.7508 |
| Brunei Darussalam | 2016 | Men | –0.82 | –1.06 | –0.58 | < 0.0001 |
| Brunei Darussalam | 2016 | Women | –0.34 | –0.55 | –0.13 | < 0.0001 |
| Chile | 2017 | Men | 0.1 | –0.39 | 0.58 | 0.0001 |
| Chile | 2017 | Women | –0.22 | –0.36 | –0.08 | < 0.0001 |
| Jordan | 2019 | Men | 0.89 | 0.31 | 1.46 | 0.0065 |
| Jordan | 2019 | Women | 0.32 | 0 | 0.64 | 0.4142 |
| Lebanon | 2017 | Men | –0.36 | –0.85 | 0.14 | 0.2074 |
| Lebanon | 2017 | Women | –0.12 | –0.45 | 0.22 | 0.1591 |
| Malawi | 2017 | Men | 0.77 | 0.39 | 1.16 | < 0.0001 |
| Malawi | 2017 | Women | 1.23 | 0.95 | 1.51 | < 0.0001 |
| Mongolia | 2013 | Men | 0.33 | –0.02 | 0.68 | 0.0184 |
| Mongolia | 2013 | Women | 0.16 | –0.03 | 0.35 | 0.2655 |
| Mongolia | 2019 | Men | 0.35 | 0.23 | 0.48 | < 0.0001 |
| Mongolia | 2019 | Women | 0.08 | –0.01 | 0.17 | 0.5155 |
| Morocco | 2017 | Men | –0.03 | –0.21 | 0.14 | 0.3083 |
| Morocco | 2017 | Women | –0.02 | –0.13 | 0.09 | 0.7259 |
| Nepal | 2019 | Men | 0.54 | 0.25 | 0.83 | < 0.0001 |
| Nepal | 2019 | Women | 0.78 | 0.61 | 0.94 | < 0.0001 |
| Solomon Islands | 2015 | Men | –0.68 | –1.26 | –0.1 | 0.0477 |
| Solomon Islands | 2015 | Women | –0.51 | –1.1 | 0.09 | 0.0539 |
| Sudan | 2016 | Men | –0.53 | –1.15 | 0.08 | 0.2111 |
| Sudan | 2016 | Women | –0.13 | –0.45 | 0.19 | 0.0674 |
| Tokelau | 2014 | Men | –0.35 | –0.53 | –0.16 | 0.2248 |
| Tokelau | 2014 | Women | –0.84 | –1.22 | –0.45 | 0.0026 |
| Tonga | 2017 | Men | –1.31 | –1.58 | –1.05 | < 0.0001 |
| Tonga | 2017 | Women | –1.22 | –1.39 | –1.05 | < 0.0001 |
| Turkmenistan | 2018 | Men | –0.44 | –0.52 | –0.36 | < 0.0001 |
| Turkmenistan | 2018 | Women | –0.45 | –0.51 | –0.39 | < 0.0001 |

*Appendix 1—table 4 Continued*

| Country | Year | Sex | Mean difference | Mean difference lower 95% confidence interval | Mean difference upper 95% confidence interval | p-Value |
|---|---|---|---|---|---|---|
| Zambia | 2017 | Men | –0.47 | –0.74 | –0.19 | < 0.0001 |
| Zambia | 2017 | Women | –0.02 | –0.21 | 0.17 | 0.3438 |

p-Value for paired *t* Student test between observed and predicted.

**Appendix 1—table 5.** Observed mean salt intake (g/day) by equation and sex in each survey included in the machine learning (ML) model development.

| Country | Year | Sex | Mean salt intake | Mean salt intake lower 95% confidence interval | Mean salt intake upper 95% confidence interval | Category |
|---|---|---|---|---|---|---|
| Armenia | 2016 | Men | 9.46 | 9.11 | 9.81 | Observed_intersalt |
| Armenia | 2016 | Men | 14.58 | 13.71 | 15.44 | Observed_kawasaki |
| Armenia | 2016 | Men | 10.21 | 9.71 | 10.7 | Observed_tanaka |
| Armenia | 2016 | Men | 12.71 | 12.19 | 13.23 | Observed_toft |
| Armenia | 2016 | Women | 7.44 | 7.26 | 7.62 | Observed_intersalt |
| Armenia | 2016 | Women | 12.48 | 11.87 | 13.09 | Observed_kawasaki |
| Armenia | 2016 | Women | 9.98 | 9.59 | 10.36 | Observed_tanaka |
| Armenia | 2016 | Women | 8.41 | 8.26 | 8.57 | Observed_toft |
| Azerbaijan | 2017 | Men | 10.39 | 10.06 | 10.72 | Observed_intersalt |
| Azerbaijan | 2017 | Men | 14.82 | 14.21 | 15.42 | Observed_kawasaki |
| Azerbaijan | 2017 | Men | 10.31 | 9.98 | 10.64 | Observed_tanaka |
| Azerbaijan | 2017 | Men | 12.81 | 12.45 | 13.18 | Observed_toft |
| Azerbaijan | 2017 | Women | 7.94 | 7.75 | 8.14 | Observed_intersalt |
| Azerbaijan | 2017 | Women | 12.65 | 12.22 | 13.08 | Observed_kawasaki |
| Azerbaijan | 2017 | Women | 10.14 | 9.87 | 10.41 | Observed_tanaka |
| Azerbaijan | 2017 | Women | 8.45 | 8.33 | 8.56 | Observed_toft |
| Bangladesh | 2018 | Men | 8.59 | 8.42 | 8.75 | Observed_intersalt |
| Bangladesh | 2018 | Men | 12.59 | 12.25 | 12.93 | Observed_kawasaki |
| Bangladesh | 2018 | Men | 8.81 | 8.62 | 9.01 | Observed_tanaka |
| Bangladesh | 2018 | Men | 11.62 | 11.4 | 11.85 | Observed_toft |
| Bangladesh | 2018 | Women | 7.27 | 7.17 | 7.37 | Observed_intersalt |
| Bangladesh | 2018 | Women | 12.09 | 11.78 | 12.4 | Observed_kawasaki |
| Bangladesh | 2018 | Women | 9 | 8.82 | 9.19 | Observed_tanaka |
| Bangladesh | 2018 | Women | 8.33 | 8.25 | 8.42 | Observed_toft |
| Belarus | 2017 | Men | 10.14 | 9.94 | 10.35 | Observed_intersalt |
| Belarus | 2017 | Men | 14.22 | 13.85 | 14.6 | Observed_kawasaki |
| Belarus | 2017 | Men | 10.16 | 9.95 | 10.38 | Observed_tanaka |
| Belarus | 2017 | Men | 12.46 | 12.24 | 12.69 | Observed_toft |

*Appendix 1—table 5 Continued on next page*

*Appendix 1—table 5 Continued*

| Country | Year | Sex | Mean salt intake | Mean salt intake lower 95% confidence interval | Mean salt intake upper 95% confidence interval | Category |
|---|---|---|---|---|---|---|
| Belarus | 2017 | Women | 7.56 | 7.41 | 7.72 | Observed_intersalt |
| Belarus | 2017 | Women | 11.43 | 11.1 | 11.75 | Observed_kawasaki |
| Belarus | 2017 | Women | 9.59 | 9.37 | 9.8 | Observed_tanaka |
| Belarus | 2017 | Women | 8.09 | 8 | 8.18 | Observed_toft |
| Bhutan | 2014 | Men | 9.58 | 9.27 | 9.88 | Observed_intersalt |
| Bhutan | 2014 | Men | 15.05 | 14.23 | 15.87 | Observed_kawasaki |
| Bhutan | 2014 | Men | 10.06 | 9.64 | 10.48 | Observed_tanaka |
| Bhutan | 2014 | Men | 13 | 12.51 | 13.49 | Observed_toft |
| Bhutan | 2014 | Women | 8.1 | 7.94 | 8.27 | Observed_intersalt |
| Bhutan | 2014 | Women | 14.24 | 13.72 | 14.76 | Observed_kawasaki |
| Bhutan | 2014 | Women | 10.54 | 10.22 | 10.86 | Observed_tanaka |
| Bhutan | 2014 | Women | 8.85 | 8.72 | 8.99 | Observed_toft |
| Bhutan | 2019 | Men | 9.1 | 8.85 | 9.35 | Observed_intersalt |
| Bhutan | 2019 | Men | 12.81 | 12.23 | 13.39 | Observed_kawasaki |
| Bhutan | 2019 | Men | 8.81 | 8.51 | 9.11 | Observed_tanaka |
| Bhutan | 2019 | Men | 11.62 | 11.28 | 11.97 | Observed_toft |
| Bhutan | 2019 | Women | 7.53 | 7.33 | 7.73 | Observed_intersalt |
| Bhutan | 2019 | Women | 11.59 | 11.22 | 11.96 | Observed_kawasaki |
| Bhutan | 2019 | Women | 8.9 | 8.67 | 9.12 | Observed_tanaka |
| Bhutan | 2019 | Women | 8.18 | 8.07 | 8.28 | Observed_toft |
| Brunei Darussalam | 2016 | Men | 8.95 | 8.66 | 9.25 | Observed_intersalt |
| Brunei Darussalam | 2016 | Men | 11.51 | 10.95 | 12.08 | Observed_kawasaki |
| Brunei Darussalam | 2016 | Men | 8.17 | 7.89 | 8.45 | Observed_tanaka |
| Brunei Darussalam | 2016 | Men | 10.79 | 10.44 | 11.14 | Observed_toft |
| Brunei Darussalam | 2016 | Women | 7.3 | 7.05 | 7.54 | Observed_intersalt |
| Brunei Darussalam | 2016 | Women | 10.52 | 10.02 | 11.01 | Observed_kawasaki |
| Brunei Darussalam | 2016 | Women | 8.38 | 8.08 | 8.69 | Observed_tanaka |
| Brunei Darussalam | 2016 | Women | 7.88 | 7.73 | 8.03 | Observed_toft |
| Chile | 2017 | Men | 9.75 | 9.18 | 10.31 | Observed_intersalt |
| Chile | 2017 | Men | 12.86 | 12.07 | 13.66 | Observed_kawasaki |
| Chile | 2017 | Men | 9.25 | 8.84 | 9.66 | Observed_tanaka |

*Appendix 1—table 5 Continued on next page*

*Appendix 1—table 5 Continued*

| Country | Year | Sex | Mean salt intake | Mean salt intake lower 95% confidence interval | Mean salt intake upper 95% confidence interval | Category |
|---|---|---|---|---|---|---|
| Chile | 2017 | Men | 11.66 | 11.14 | 12.17 | Observed_toft |
| Chile | 2017 | Women | 7.64 | 7.45 | 7.83 | Observed_intersalt |
| Chile | 2017 | Women | 11.11 | 10.81 | 11.4 | Observed_kawasaki |
| Chile | 2017 | Women | 9.13 | 8.93 | 9.32 | Observed_tanaka |
| Chile | 2017 | Women | 8.06 | 7.97 | 8.15 | Observed_toft |
| Jordan | 2019 | Men | 10.2 | 9.52 | 10.88 | Observed_intersalt |
| Jordan | 2019 | Men | 13.98 | 12.73 | 15.23 | Observed_kawasaki |
| Jordan | 2019 | Men | 9.84 | 9.17 | 10.51 | Observed_tanaka |
| Jordan | 2019 | Men | 12.29 | 11.56 | 13.02 | Observed_toft |
| Jordan | 2019 | Women | 8.1 | 7.75 | 8.45 | Observed_intersalt |
| Jordan | 2019 | Women | 12.1 | 11.48 | 12.72 | Observed_kawasaki |
| Jordan | 2019 | Women | 9.74 | 9.34 | 10.13 | Observed_tanaka |
| Jordan | 2019 | Women | 8.34 | 8.17 | 8.5 | Observed_toft |
| Lebanon | 2017 | Men | 9.53 | 9.06 | 9.99 | Observed_intersalt |
| Lebanon | 2017 | Men | 12.72 | 11.65 | 13.79 | Observed_kawasaki |
| Lebanon | 2017 | Men | 9.22 | 8.61 | 9.84 | Observed_tanaka |
| Lebanon | 2017 | Men | 11.48 | 10.82 | 12.14 | Observed_toft |
| Lebanon | 2017 | Women | 7.51 | 7.07 | 7.95 | Observed_intersalt |
| Lebanon | 2017 | Women | 11.35 | 10.45 | 12.25 | Observed_kawasaki |
| Lebanon | 2017 | Women | 9.37 | 8.75 | 10 | Observed_tanaka |
| Lebanon | 2017 | Women | 8.03 | 7.76 | 8.3 | Observed_toft |
| Malawi | 2017 | Men | 9.54 | 9.16 | 9.91 | Observed_intersalt |
| Malawi | 2017 | Men | 14.02 | 13.4 | 14.64 | Observed_kawasaki |
| Malawi | 2017 | Men | 9.43 | 9.08 | 9.77 | Observed_tanaka |
| Malawi | 2017 | Men | 12.4 | 12.04 | 12.77 | Observed_toft |
| Malawi | 2017 | Women | 8.34 | 8.03 | 8.64 | Observed_intersalt |
| Malawi | 2017 | Women | 13.43 | 12.76 | 14.11 | Observed_kawasaki |
| Malawi | 2017 | Women | 10.17 | 9.75 | 10.58 | Observed_tanaka |
| Malawi | 2017 | Women | 8.64 | 8.47 | 8.82 | Observed_toft |
| Mongolia | 2013 | Men | 9.83 | 9.34 | 10.32 | Observed_intersalt |
| Mongolia | 2013 | Men | 13.37 | 12.74 | 14.01 | Observed_kawasaki |
| Mongolia | 2013 | Men | 9.48 | 9.13 | 9.83 | Observed_tanaka |
| Mongolia | 2013 | Men | 12.04 | 11.64 | 12.45 | Observed_toft |
| Mongolia | 2013 | Women | 7.79 | 7.6 | 7.97 | Observed_intersalt |
| Mongolia | 2013 | Women | 11.92 | 11.34 | 12.5 | Observed_kawasaki |
| Mongolia | 2013 | Women | 9.54 | 9.16 | 9.92 | Observed_tanaka |

*Appendix 1—table 5 Continued on next page*

*Appendix 1—table 5 Continued*

| Country | Year | Sex | Mean salt intake | Mean salt intake lower 95% confidence interval | Mean salt intake upper 95% confidence interval | Category |
|---|---|---|---|---|---|---|
| Mongolia | 2013 | Women | 8.24 | 8.08 | 8.4 | Observed_toft |
| Mongolia | 2019 | Men | 9.68 | 9.5 | 9.85 | Observed_intersalt |
| Mongolia | 2019 | Men | 14.83 | 14.49 | 15.17 | Observed_kawasaki |
| Mongolia | 2019 | Men | 10.14 | 9.95 | 10.32 | Observed_tanaka |
| Mongolia | 2019 | Men | 12.84 | 12.64 | 13.05 | Observed_toft |
| Mongolia | 2019 | Women | 7.46 | 7.34 | 7.59 | Observed_intersalt |
| Mongolia | 2019 | Women | 12.13 | 11.81 | 12.44 | Observed_kawasaki |
| Mongolia | 2019 | Women | 9.63 | 9.43 | 9.84 | Observed_tanaka |
| Mongolia | 2019 | Women | 8.31 | 8.23 | 8.4 | Observed_toft |
| Morocco | 2017 | Men | 9.03 | 8.82 | 9.24 | Observed_intersalt |
| Morocco | 2017 | Men | 13.04 | 12.63 | 13.44 | Observed_kawasaki |
| Morocco | 2017 | Men | 9.33 | 9.1 | 9.56 | Observed_tanaka |
| Morocco | 2017 | Men | 11.75 | 11.5 | 12 | Observed_toft |
| Morocco | 2017 | Women | 7.47 | 7.35 | 7.59 | Observed_intersalt |
| Morocco | 2017 | Women | 11.72 | 11.41 | 12.04 | Observed_kawasaki |
| Morocco | 2017 | Women | 9.48 | 9.28 | 9.68 | Observed_tanaka |
| Morocco | 2017 | Women | 8.18 | 8.09 | 8.26 | Observed_toft |
| Nepal | 2019 | Men | 9.55 | 9.22 | 9.87 | Observed_intersalt |
| Nepal | 2019 | Men | 16.6 | 15.92 | 17.27 | Observed_kawasaki |
| Nepal | 2019 | Men | 10.69 | 10.33 | 11.04 | Observed_tanaka |
| Nepal | 2019 | Men | 14.04 | 13.64 | 14.44 | Observed_toft |
| Nepal | 2019 | Women | 7.84 | 7.65 | 8.04 | Observed_intersalt |
| Nepal | 2019 | Women | 15.35 | 14.82 | 15.88 | Observed_kawasaki |
| Nepal | 2019 | Women | 10.9 | 10.57 | 11.24 | Observed_tanaka |
| Nepal | 2019 | Women | 9.12 | 8.99 | 9.25 | Observed_toft |
| Solomon Islands | 2015 | Men | 8.74 | 8.08 | 9.4 | Observed_intersalt |
| Solomon Islands | 2015 | Men | 12.99 | 11.06 | 14.93 | Observed_kawasaki |
| Solomon Islands | 2015 | Men | 8.87 | 7.97 | 9.77 | Observed_tanaka |
| Solomon Islands | 2015 | Men | 11.62 | 10.43 | 12.8 | Observed_toft |
| Solomon Islands | 2015 | Women | 7.03 | 6.42 | 7.64 | Observed_intersalt |
| Solomon Islands | 2015 | Women | 11.38 | 8.78 | 13.98 | Observed_kawasaki |
| Solomon Islands | 2015 | Women | 8.98 | 7.34 | 10.61 | Observed_tanaka |

*Appendix 1—table 5 Continued on next page*

*Appendix 1—table 5 Continued*

| Country | Year | Sex | Mean salt intake | Mean salt intake lower 95% confidence interval | Mean salt intake upper 95% confidence interval | Category |
|---|---|---|---|---|---|---|
| Solomon Islands | 2015 | Women | 7.95 | 7.26 | 8.64 | Observed_toft |
| Sudan | 2016 | Men | 8.53 | 7.73 | 9.33 | Observed_intersalt |
| Sudan | 2016 | Men | 11.66 | 10.66 | 12.66 | Observed_kawasaki |
| Sudan | 2016 | Men | 8.49 | 7.91 | 9.08 | Observed_tanaka |
| Sudan | 2016 | Men | 10.83 | 10.17 | 11.5 | Observed_toft |
| Sudan | 2016 | Women | 7.49 | 7.09 | 7.88 | Observed_intersalt |
| Sudan | 2016 | Women | 11.3 | 10.6 | 12.01 | Observed_kawasaki |
| Sudan | 2016 | Women | 9.31 | 8.85 | 9.78 | Observed_tanaka |
| Sudan | 2016 | Women | 8.09 | 7.89 | 8.3 | Observed_toft |
| Tokelau | 2014 | Men | 10.29 | 10.18 | 10.4 | Observed_intersalt |
| Tokelau | 2014 | Men | 14.33 | 13.16 | 15.5 | Observed_kawasaki |
| Tokelau | 2014 | Men | 10.1 | 9.48 | 10.72 | Observed_tanaka |
| Tokelau | 2014 | Men | 12.42 | 11.71 | 13.14 | Observed_toft |
| Tokelau | 2014 | Women | 8.12 | 7.61 | 8.63 | Observed_intersalt |
| Tokelau | 2014 | Women | 11.4 | 9.85 | 12.95 | Observed_kawasaki |
| Tokelau | 2014 | Women | 9.71 | 8.69 | 10.72 | Observed_tanaka |
| Tokelau | 2014 | Women | 8.15 | 7.76 | 8.54 | Observed_toft |
| Tonga | 2017 | Men | 9.19 | 8.89 | 9.48 | Observed_intersalt |
| Tonga | 2017 | Men | 10.06 | 9.17 | 10.95 | Observed_kawasaki |
| Tonga | 2017 | Men | 7.72 | 7.22 | 8.22 | Observed_tanaka |
| Tonga | 2017 | Men | 9.77 | 9.19 | 10.35 | Observed_toft |
| Tonga | 2017 | Women | 7.63 | 7.45 | 7.81 | Observed_intersalt |
| Tonga | 2017 | Women | 9.37 | 8.88 | 9.87 | Observed_kawasaki |
| Tonga | 2017 | Women | 8.41 | 8.06 | 8.76 | Observed_tanaka |
| Tonga | 2017 | Women | 7.53 | 7.37 | 7.68 | Observed_toft |
| Turkmenistan | 2018 | Men | 8.94 | 8.79 | 9.09 | Observed_intersalt |
| Turkmenistan | 2018 | Men | 12.11 | 11.93 | 12.3 | Observed_kawasaki |
| Turkmenistan | 2018 | Men | 8.85 | 8.74 | 8.96 | Observed_tanaka |
| Turkmenistan | 2018 | Men | 11.2 | 11.09 | 11.32 | Observed_toft |
| Turkmenistan | 2018 | Women | 6.76 | 6.68 | 6.83 | Observed_intersalt |
| Turkmenistan | 2018 | Women | 10.1 | 9.93 | 10.26 | Observed_kawasaki |
| Turkmenistan | 2018 | Women | 8.53 | 8.41 | 8.65 | Observed_tanaka |
| Turkmenistan | 2018 | Women | 7.78 | 7.73 | 7.83 | Observed_toft |
| Zambia | 2017 | Men | 8.45 | 8.15 | 8.75 | Observed_intersalt |
| Zambia | 2017 | Men | 12.7 | 12.09 | 13.3 | Observed_kawasaki |

*Appendix 1—table 5 Continued on next page*

*Appendix 1—table 5 Continued*

| Country | Year | Sex | Mean salt intake | Mean salt intake lower 95% confidence interval | Mean salt intake upper 95% confidence interval | Category |
|---|---|---|---|---|---|---|
| Zambia | 2017 | Men | 8.8 | 8.46 | 9.13 | Observed_tanaka |
| Zambia | 2017 | Men | 11.48 | 11.11 | 11.86 | Observed_toft |
| Zambia | 2017 | Women | 7.01 | 6.81 | 7.22 | Observed_intersalt |
| Zambia | 2017 | Women | 11.11 | 10.66 | 11.56 | Observed_kawasaki |
| Zambia | 2017 | Women | 8.8 | 8.52 | 9.09 | Observed_tanaka |
| Zambia | 2017 | Women | 8 | 7.86 | 8.13 | Observed_toft |

**Appendix 1—table 6.** Weighted distribution of predictors in each of the 54 national surveys included in the application of the model herein developed.

| Country | Year | Sample size | Mean age (years) | Age range (years) | Proportion of men (%) | Mean, minimum, and maximum values of SBP (mmHg) | Mean, minimum, and maximum values of DBP (mmHg) | Mean, minimum, and maximum values of weight (kg) | Mean, minimum, and maximum values of height (m) |
|---|---|---|---|---|---|---|---|---|---|
| American Samoa | 2004 | 2043 | 40 | 25–64 | 50.3 | 131 (84–230) | 82 (46–134) | 100.4 (38.6–219.1) | 1.69 (1.36–2.19) |
| Benin | 2015 | 4841 | 34 | 18–69 | 49.6 | 126 (74–254) | 82 (45–142) | 62.3 (30–167) | 1.64 (1.21–1.98) |
| Bahamas | 2012 | 1400 | 42 | 24–64 | 49.9 | 127 (73–248) | 82 (32–140) | 84.8 (27.9–184.9) | 1.67 (1.15–2.03) |
| Barbados | 2007 | 282 | 43 | 25–69 | 51.9 | 122 (86–191) | 80 (55–115) | 77.5 (40.6–232.1) | 1.67 (1.17–1.93) |
| British Virgin Islands | 2009 | 1067 | 43 | 25–64 | 54.1 | 130 (81–226) | 80 (48–126) | 83.2 (39.6–176.9) | 1.7 (1.14–2.26) |
| Botswana | 2014 | 3894 | 33 | 15–69 | 52.1 | 128 (84–262) | 80 (47–148) | 63.9 (31.7–171.1) | 1.66 (1.02–2) |
| Cook Islands | 2015 | 879 | 39 | 18–64 | 46.5 | 128 (92–194) | 79 (45–118) | 98.6 (49.1–205.1) | 1.69 (1.07–1.96) |
| Comoros | 2011 | 5029 | 39 | 25–64 | 52.6 | 128 (82–236) | 79 (48–144) | 64.2 (23.5–166) | 1.61 (1–2.15) |
| Cabo Verde | 2007 | 1723 | 38 | 25–64 | 50.3 | 133 (86–234) | 80 (48–140) | 68.3 (35–150) | 1.68 (1.23–1.96) |
| Cayman Islands | 2012 | 1229 | 42 | 24–64 | 50.7 | 125 (84–208) | 76 (46–127) | 82.3 (31–196) | 1.69 (1–2.1) |
| Algeria | 2017 | 6536 | 38 | 18–69 | 51.7 | 127 (77–227) | 75 (32–137) | 73.3 (25–174) | 1.67 (1.02–2.05) |
| Ecuador | 2018 | 4466 | 40 | 18–69 | 49.4 | 120 (78–220) | 76 (42–130) | 69.2 (33.4–198.4) | 1.59 (1.24–1.93) |
| Eritrea | 2010 | 5651 | 42 | 25–69 | 17.2 | 117 (72–230) | 74 (46–130) | 51.8 (28.1–99.1) | 1.6 (1.16–1.89) |
| Ethiopia | 2015 | 9270 | 31 | 15–69 | 56.1 | 120 (71–250) | 78 (30–142) | 54.4 (20–99.5) | 1.63 (1.05–2) |
| Fiji | 2011 | 2492 | 42 | 25–64 | 51 | 130 (84–228) | 80 (39–143) | 78.6 (30.3–198.1) | 1.68 (1.03–1.94) |
| Gambia | 2010 | 3496 | 38 | 25–64 | 50.4 | 130 (85–252) | 80 (44–144) | 64.8 (26.5–168.9) | 1.64 (1–2) |
| Grenada | 2011 | 1055 | 41 | 25–64 | 50.7 | 131 (71–212) | 80 (50–128) | 77.6 (40.8–158.8) | 1.7 (1.32–2.49) |
| Guyana | 2016 | 2625 | 37 | 18–69 | 52 | 126 (74–245) | 78 (37–149) | 69.9 (26.4–198) | 1.63 (1.01–2.07) |
| Iraq | 2015 | 3655 | 35 | 18–69 | 53.6 | 128 (78–225) | 83 (45–150) | 76.5 (36.6–187.2) | 1.65 (1.01–1.97) |
| Kenya | 2015 | 4270 | 34 | 16–69 | 50.6 | 125 (76–262) | 81 (46–146) | 63.2 (30–171.3) | 1.65 (1.01–1.95) |
| Kyrgyzstan | 2013 | 2539 | 41 | 25–64 | 51.9 | 133 (82–244) | 87 (56–150) | 71.7 (36.6–162.4) | 1.64 (1.38–1.95) |
| Cambodia | 2010 | 5223 | 40 | 25–64 | 49.4 | 116 (70–226) | 72 (42–138) | 53.7 (21.1–111) | 1.57 (1.24–1.85) |
| Kiribati | 2016 | 1240 | 40 | 18–69 | 42.8 | 128 (85–220) | 85 (49–148) | 81.1 (30–219) | 1.64 (1.22–1.89) |
| Kuwait | 2014 | 2871 | 36 | 18–69 | 49.5 | 120 (70–240) | 77 (50–130) | 80.5 (37.3–195) | 1.65 (1.04–1.96) |

*Appendix 1—table 6 Continued on next page*

*Appendix 1—table 6 Continued*

| Country | Year | Sample size | Mean age (years) | Age range (years) | Proportion of men (%) | Mean, minimum, and maximum values of SBP (mmHg) | Mean, minimum, and maximum values of DBP (mmHg) | Mean, minimum, and maximum values of weight (kg) | Mean, minimum, and maximum values of height (m) |
|---|---|---|---|---|---|---|---|---|---|
| Lao People's Democratic Republic | 2013 | 2464 | 39 | 16–65 | 42.3 | 119 (72–240) | 76 (30–130) | 54.2 (27–103.1) | 1.54 (1.16–1.97) |
| Liberia | 2011 | 2242 | 40 | 25–64 | 50.7 | 129 (88–232) | 80 (32–138) | 65.4 (32–163) | 1.58 (1–2.5) |
| Libya | 2009 | 3223 | 37 | 25–64 | 51.5 | 133 (74–238) | 79 (44–148) | 77 (31.7–186.2) | 1.67 (1–1.97) |
| Sri Lanka | 2015 | 4566 | 39 | 18–69 | 51.5 | 125 (74–258) | 81 (36–150) | 58 (26.2–156.9) | 1.59 (1.02–1.9) |
| Lesotho | 2012 | 2162 | 38 | 25–64 | 49.8 | 126 (78–250) | 83 (46–146) | 66.2 (21.5–164.6) | 1.61 (1.02–1.97) |
| Republic of Moldova | 2013 | 4077 | 39 | 18–69 | 52.5 | 133 (83–257) | 85 (49–148) | 75 (32.5–166) | 1.68 (1.2–1.98) |
| Marshall Islands | 2018 | 2657 | 39 | 17–69 | 48.5 | 120 (70–220) | 75 (40–134) | 74.4 (27–226.5) | 1.58 (1.01–2.15) |
| Myanmar | 2014 | 7892 | 42 | 25–64 | 50.4 | 126 (70–252) | 82 (35–144) | 57.1 (26.3–173) | 1.59 (1–2.18) |
| Mozambique | 2005 | 723 | 41 | 24–64 | 46 | 139 (85–220) | 82 (46–143) | 56.7 (33.4–109.5) | 1.6 (1.02–1.89) |
| Namibia | 2005 | 752 | 41 | 25–64 | 41.3 | 137 (87–230) | 86 (50–132) | 63.7 (26.5–134.3) | 1.63 (1.12–2) |
| Niger | 2007 | 2638 | 37 | 15–64 | 54.1 | 134 (70–260) | 82 (40–145) | 59.5 (24.3–162.2) | 1.67 (1.01–2.1) |
| Niue | 2012 | 779 | 40 | 15–69 | 50.1 | 128 (89–223) | 76 (44–117) | 91.5 (44.7–165.9) | 1.69 (1.17–1.96) |
| Nauru | 2016 | 1037 | 36 | 18–69 | 50 | 123 (76–223) | 80 (46–125) | 92.4 (43.4–197.9) | 1.63 (1.41–1.86) |
| Palau | 2013 | 2148 | 43 | 25–64 | 53 | 138 (87–236) | 85 (40–135) | 79.4 (32–180.6) | 1.62 (1.02–2.03) |
| French Polynesia | 2010 | 2239 | 36 | 18–64 | 50.7 | 125 (86–230) | 79 (48–150) | 86.2 (41–193) | 1.7 (1.41–2) |
| Qatar | 2012 | 2287 | 35 | 18–64 | 50.9 | 119 (78–203) | 79 (46–130) | 79.1 (34.4–190.5) | 1.64 (1.35–2) |
| Rwanda | 2013 | 6882 | 32 | 15–64 | 48.8 | 121 (75–250) | 78 (45–140) | 57 (23.1–165.8) | 1.6 (1–1.91) |
| Sierra Leone | 2009 | 4473 | 40 | 25–64 | 50.3 | 131 (72–220) | 81 (42–148) | 60 (28–185) | 1.62 (1–2.34) |
| Sao Tome and Principe | 2008 | 2272 | 40 | 25–64 | 48.4 | 135 (78–240) | 82 (34–143) | 66.1 (30–186.2) | 1.64 (1.01–1.98) |
| Eswatini | 2014 | 3042 | 31 | 15–69 | 47.4 | 124 (72–252) | 80 (42–150) | 67.8 (22.2–227.6) | 1.63 (1.01–2.02) |
| Togo | 2011 | 3995 | 32 | 15–64 | 49.3 | 123 (70–251) | 77 (31–142) | 61.6 (26–165) | 1.64 (1.02–1.99) |
| Tajikistan | 2017 | 2643 | 32 | 18–69 | 53.8 | 129 (81–267) | 84 (54–150) | 66.7 (27.8–148) | 1.63 (1.09–2) |
| Timor-Leste | 2014 | 2480 | 36 | 18–69 | 63.8 | 130 (72–235) | 84 (42–136) | 52 (27–165) | 1.57 (1.24–1.83) |
| Tuvalu | 2015 | 1024 | 39 | 18–69 | 54.9 | 134 (92–246) | 84 (48–145) | 91.9 (35.8–181.8) | 1.68 (1.17–2.06) |
| United Republic of Tanzania | 2012 | 5381 | 39 | 25–64 | 50.6 | 129 (80–240) | 80 (40–146) | 60.6 (29–171.1) | 1.63 (1.13–1.97) |
| Uganda | 2014 | 3673 | 35 | 18–69 | 50.5 | 125 (83–249) | 81 (50–148) | 59.4 (30.2–165) | 1.62 (1.15–2.03) |
| Uruguay | 2014 | 2207 | 38 | 15–64 | 47.8 | 125 (82–232) | 79 (44–134) | 74.6 (34.3–158) | 1.67 (1.36–2.05) |
| Vietnam | 2015 | 3033 | 39 | 18–69 | 50.4 | 120 (71–224) | 77 (40–128) | 54.7 (27.8–106.4) | 1.58 (1.01–1.98) |
| Vanuatu | 2011 | 4420 | 40 | 25–64 | 47.7 | 130 (77–269) | 80 (38–139) | 69.4 (28.3–199.8) | 1.63 (1.02–2.1) |
| Samoa | 2013 | 1490 | 37 | 18–64 | 54.1 | 125 (80–222) | 75 (44–132) | 90.3 (32.1–160) | 1.68 (1.22–1.97) |

SBP: systolic blood pressure; DBP: diastolic blood pressure.

**Appendix 1—table 7.** Predicted mean salt intake (g/day) by sex in each of the 54 national surveys included in the application of the model herein developed.

| Country | Year | Sex | Mean salt intake | Mean salt intake lower 95% confidence interval | Mean salt intake upper 95% confidence interval |
|---|---|---|---|---|---|
| Algeria | 2017 | Men | 9.26 | 9.22 | 9.3 |
| Algeria | 2017 | Women | 7.54 | 7.5 | 7.58 |
| Algeria | 2017 | Total | 8.43 | 8.39 | 8.47 |
| American Samoa | 2004 | Men | 10.9 | 10.8 | 10.99 |
| American Samoa | 2004 | Women | 9.03 | 8.96 | 9.11 |
| American Samoa | 2004 | Total | 9.97 | 9.93 | 10.01 |
| Bahamas | 2012 | Men | 10.09 | 9.83 | 10.35 |
| Bahamas | 2012 | Women | 8.11 | 7.81 | 8.4 |
| Bahamas | 2012 | Total | 9.1 | 8.9 | 9.29 |
| Barbados | 2007 | Men | 9.42 | 9.25 | 9.6 |
| Barbados | 2007 | Women | 7.85 | 7.54 | 8.17 |
| Barbados | 2007 | Total | 8.67 | 8.45 | 8.89 |
| Benin | 2015 | Men | 8.96 | 8.9 | 9.03 |
| Benin | 2015 | Women | 7.01 | 6.89 | 7.12 |
| Benin | 2015 | Total | 7.98 | 7.81 | 8.15 |
| Botswana | 2014 | Men | 8.74 | 8.68 | 8.79 |
| Botswana | 2014 | Women | 7.2 | 7.14 | 7.26 |
| Botswana | 2014 | Total | 8 | 7.94 | 8.06 |
| British Virgin Islands | 2009 | Men | 9.73 | 9.66 | 9.81 |
| British Virgin Islands | 2009 | Women | 7.85 | 7.82 | 7.88 |
| British Virgin Islands | 2009 | Total | 8.87 | 8.82 | 8.92 |
| Cabo Verde | 2007 | Men | 8.98 | 8.93 | 9.03 |
| Cabo Verde | 2007 | Women | 7.13 | 7.03 | 7.23 |
| Cabo Verde | 2007 | Total | 8.06 | 7.97 | 8.16 |
| Cambodia | 2010 | Men | 8.83 | 8.8 | 8.86 |
| Cambodia | 2010 | Women | 6.83 | 6.81 | 6.86 |
| Cambodia | 2010 | Total | 7.82 | 7.78 | 7.86 |
| Cayman Islands | 2012 | Men | 9.73 | 9.69 | 9.77 |
| Cayman Islands | 2012 | Women | 7.92 | 7.61 | 8.23 |
| Cayman Islands | 2012 | Total | 8.84 | 8.75 | 8.92 |
| Comoros | 2011 | Men | 9.06 | 9.02 | 9.1 |
| Comoros | 2011 | Women | 7.43 | 7.38 | 7.47 |
| Comoros | 2011 | Total | 8.29 | 8.24 | 8.33 |
| Cook Islands | 2015 | Men | 10.87 | 10.73 | 11.01 |
| Cook Islands | 2015 | Women | 8.74 | 8.63 | 8.86 |

*Appendix 1—table 7 Continued on next page*

*Appendix 1—table 7 Continued*

| Country | Year | Sex | Mean salt intake | Mean salt intake lower 95% confidence interval | Mean salt intake upper 95% confidence interval |
|---|---|---|---|---|---|
| Cook Islands | 2015 | Total | 9.73 | 9.59 | 9.88 |
| Ecuador | 2018 | Men | 9.6 | 9.55 | 9.65 |
| Ecuador | 2018 | Women | 7.65 | 7.6 | 7.69 |
| Ecuador | 2018 | Total | 8.61 | 8.55 | 8.68 |
| Eritrea | 2010 | Men | 8.32 | 8.27 | 8.37 |
| Eritrea | 2010 | Women | 6.48 | 6.43 | 6.52 |
| Eritrea | 2010 | Total | 6.79 | 6.75 | 6.84 |
| Eswatini | 2014 | Men | 9.11 | 9.02 | 9.2 |
| Eswatini | 2014 | Women | 7.62 | 7.56 | 7.68 |
| Eswatini | 2014 | Total | 8.33 | 8.27 | 8.39 |
| Ethiopia | 2015 | Men | 8.52 | 8.49 | 8.54 |
| Ethiopia | 2015 | Women | 6.62 | 6.59 | 6.65 |
| Ethiopia | 2015 | Total | 7.68 | 7.65 | 7.72 |
| Fiji | 2011 | Men | 9.53 | 9.44 | 9.62 |
| Fiji | 2011 | Women | 7.84 | 7.76 | 7.91 |
| Fiji | 2011 | Total | 8.7 | 8.6 | 8.8 |
| French Polynesia | 2010 | Men | 10.1 | 10 | 10.2 |
| French Polynesia | 2010 | Women | 8 | 7.9 | 8.1 |
| French Polynesia | 2010 | Total | 9.06 | 8.98 | 9.15 |
| Gambia | 2010 | Men | 9.05 | 8.94 | 9.17 |
| Gambia | 2010 | Women | 7.17 | 7.1 | 7.25 |
| Gambia | 2010 | Total | 8.12 | 8.03 | 8.22 |
| Grenada | 2011 | Men | 9.21 | 9.12 | 9.31 |
| Grenada | 2011 | Women | 7.74 | 7.64 | 7.84 |
| Grenada | 2011 | Total | 8.49 | 8.4 | 8.58 |
| Guyana | 2016 | Men | 9.26 | 9.16 | 9.35 |
| Guyana | 2016 | Women | 7.67 | 7.6 | 7.74 |
| Guyana | 2016 | Total | 8.5 | 8.43 | 8.56 |
| Iraq | 2015 | Men | 9.66 | 9.58 | 9.75 |
| Iraq | 2015 | Women | 7.94 | 7.88 | 8.01 |
| Iraq | 2015 | Total | 8.87 | 8.8 | 8.93 |
| Kenya | 2015 | Men | 8.82 | 8.73 | 8.9 |
| Kenya | 2015 | Women | 7.13 | 7.04 | 7.21 |
| Kenya | 2015 | Total | 7.98 | 7.89 | 8.07 |
| Kiribati | 2016 | Men | 9.92 | 9.74 | 10.09 |
| Kiribati | 2016 | Women | 8.27 | 8.14 | 8.39 |

*Appendix 1—table 7 Continued on next page*

*Appendix 1—table 7 Continued*

| Country | Year | Sex | Mean salt intake | Mean salt intake lower 95% confidence interval | Mean salt intake upper 95% confidence interval |
|---|---|---|---|---|---|
| Kiribati | 2016 | Total | 8.97 | 8.86 | 9.09 |
| Kuwait | 2014 | Men | 10.06 | 9.99 | 10.12 |
| Kuwait | 2014 | Women | 7.95 | 7.91 | 8 |
| Kuwait | 2014 | Total | 8.99 | 8.94 | 9.05 |
| Kyrgyzstan | 2013 | Men | 9.45 | 9.34 | 9.55 |
| Kyrgyzstan | 2013 | Women | 7.62 | 7.56 | 7.67 |
| Kyrgyzstan | 2013 | Total | 8.57 | 8.5 | 8.63 |
| Lao People's Democratic Republic | 2013 | Men | 9.03 | 8.98 | 9.08 |
| Lao People's Democratic Republic | 2013 | Women | 7.07 | 7.02 | 7.12 |
| Lao People's Democratic Republic | 2013 | Total | 7.9 | 7.83 | 7.97 |
| Lesotho | 2012 | Men | 9.08 | 8.99 | 9.17 |
| Lesotho | 2012 | Women | 7.7 | 7.6 | 7.79 |
| Lesotho | 2012 | Total | 8.38 | 8.31 | 8.46 |
| Liberia | 2011 | Men | 9.43 | 9.32 | 9.55 |
| Liberia | 2011 | Women | 7.58 | 7.48 | 7.69 |
| Liberia | 2011 | Total | 8.52 | 8.41 | 8.63 |
| Libya | 2009 | Men | 9.51 | 9.44 | 9.59 |
| Libya | 2009 | Women | 7.81 | 7.73 | 7.89 |
| Libya | 2009 | Total | 8.69 | 8.63 | 8.75 |
| Marshall Islands | 2018 | Men | 9.92 | 9.86 | 9.99 |
| Marshall Islands | 2018 | Women | 8.16 | 8.1 | 8.21 |
| Marshall Islands | 2018 | Total | 9.01 | 8.96 | 9.07 |
| Mozambique | 2005 | Men | 8.72 | 8.62 | 8.83 |
| Mozambique | 2005 | Women | 6.92 | 6.84 | 7 |
| Mozambique | 2005 | Total | 7.75 | 7.63 | 7.87 |
| Myanmar | 2014 | Men | 8.81 | 8.74 | 8.88 |
| Myanmar | 2014 | Women | 7.07 | 6.97 | 7.17 |
| Myanmar | 2014 | Total | 7.95 | 7.88 | 8.02 |
| Namibia | 2005 | Men | 8.74 | 8.59 | 8.89 |
| Namibia | 2005 | Women | 7.24 | 6.93 | 7.56 |
| Namibia | 2005 | Total | 7.86 | 7.63 | 8.09 |
| Nauru | 2016 | Men | 10.98 | 10.87 | 11.1 |
| Nauru | 2016 | Women | 8.79 | 8.63 | 8.94 |

*Appendix 1—table 7 Continued on next page*

*Appendix 1—table 7 Continued*

| Country | Year | Sex | Mean salt intake | Mean salt intake lower 95% confidence interval | Mean salt intake upper 95% confidence interval |
|---|---|---|---|---|---|
| Nauru | 2016 | Total | 9.89 | 9.74 | 10.03 |
| Niger | 2007 | Men | 8.56 | 8.52 | 8.6 |
| Niger | 2007 | Women | 6.67 | 6.63 | 6.71 |
| Niger | 2007 | Total | 7.69 | 7.65 | 7.74 |
| Niue | 2012 | Men | 10.39 | 10.28 | 10.51 |
| Niue | 2012 | Women | 8.39 | 8.27 | 8.51 |
| Niue | 2012 | Total | 9.4 | 9.29 | 9.5 |
| Palau | 2013 | Men | 10.18 | 10.07 | 10.28 |
| Palau | 2013 | Women | 7.99 | 7.9 | 8.08 |
| Palau | 2013 | Total | 9.15 | 9.05 | 9.25 |
| Qatar | 2012 | Men | 10.02 | 9.93 | 10.11 |
| Qatar | 2012 | Women | 7.94 | 7.85 | 8.04 |
| Qatar | 2012 | Total | 9 | 8.9 | 9.09 |
| Republic of Moldova | 2013 | Men | 9.51 | 9.45 | 9.57 |
| Republic of Moldova | 2013 | Women | 7.46 | 7.41 | 7.52 |
| Republic of Moldova | 2013 | Total | 8.54 | 8.48 | 8.6 |
| Rwanda | 2013 | Men | 8.87 | 8.85 | 8.9 |
| Rwanda | 2013 | Women | 7.02 | 6.99 | 7.05 |
| Rwanda | 2013 | Total | 7.92 | 7.89 | 7.96 |
| Samoa | 2013 | Men | 10.23 | 10.09 | 10.37 |
| Samoa | 2013 | Women | 8.61 | 8.51 | 8.71 |
| Samoa | 2013 | Total | 9.49 | 9.41 | 9.57 |
| Sao Tome and Principe | 2008 | Men | 9.05 | 8.97 | 9.12 |
| Sao Tome and Principe | 2008 | Women | 7.21 | 7.1 | 7.32 |
| Sao Tome and Principe | 2008 | Total | 8.1 | 7.99 | 8.2 |
| Sierra Leone | 2009 | Men | 8.85 | 8.76 | 8.94 |
| Sierra Leone | 2009 | Women | 7 | 6.9 | 7.11 |
| Sierra Leone | 2009 | Total | 7.93 | 7.82 | 8.04 |
| Sri Lanka | 2015 | Men | 8.91 | 8.86 | 8.95 |
| Sri Lanka | 2015 | Women | 7.07 | 7.03 | 7.1 |
| Sri Lanka | 2015 | Total | 8.01 | 7.97 | 8.06 |
| Tajikistan | 2017 | Men | 9.41 | 9.34 | 9.49 |
| Tajikistan | 2017 | Women | 7.35 | 7.3 | 7.41 |

*Appendix 1—table 7 Continued on next page*

*Appendix 1—table 7 Continued*

| Country | Year | Sex | Mean salt intake | Mean salt intake lower 95% confidence interval | Mean salt intake upper 95% confidence interval |
|---|---|---|---|---|---|
| Tajikistan | 2017 | Total | 8.46 | 8.38 | 8.55 |
| Timor-Leste | 2014 | Men | 8.91 | 8.79 | 9.02 |
| Timor-Leste | 2014 | Women | 6.8 | 6.75 | 6.86 |
| Timor-Leste | 2014 | Total | 8.15 | 7.86 | 8.43 |
| Togo | 2011 | Men | 8.82 | 8.79 | 8.86 |
| Togo | 2011 | Women | 7.01 | 6.96 | 7.06 |
| Togo | 2011 | Total | 7.9 | 7.85 | 7.96 |
| Tuvalu | 2015 | Men | 10.37 | 10.24 | 10.5 |
| Tuvalu | 2015 | Women | 8.72 | 8.62 | 8.83 |
| Tuvalu | 2015 | Total | 9.63 | 9.53 | 9.73 |
| Uganda | 2014 | Men | 8.8 | 8.76 | 8.84 |
| Uganda | 2014 | Women | 7.02 | 6.96 | 7.07 |
| Uganda | 2014 | Total | 7.92 | 7.86 | 7.98 |
| United Republic of Tanzania | 2012 | Men | 8.71 | 8.63 | 8.79 |
| United Republic of Tanzania | 2012 | Women | 7.13 | 7.05 | 7.21 |
| United Republic of Tanzania | 2012 | Total | 7.93 | 7.88 | 7.98 |
| Uruguay | 2014 | Men | 9.55 | 9.48 | 9.63 |
| Uruguay | 2014 | Women | 7.47 | 7.41 | 7.52 |
| Uruguay | 2014 | Total | 8.46 | 8.39 | 8.53 |
| Vanuatu | 2011 | Men | 9.38 | 9.33 | 9.43 |
| Vanuatu | 2011 | Women | 7.45 | 7.4 | 7.5 |
| Vanuatu | 2011 | Total | 8.37 | 8.31 | 8.43 |
| Vietnam | 2015 | Men | 8.91 | 8.86 | 8.95 |
| Vietnam | 2015 | Women | 6.84 | 6.81 | 6.88 |
| Vietnam | 2015 | Total | 7.88 | 7.83 | 7.94 |

**Appendix 1—table 8.** Comparison between mean salt intake (g/day) predictions and global estimates across national surveys included in the application of our machine learning model.

| Country | Year (machine learning predictions) | Machine learning predicted mean salt intake and 95% confidence interval | Year (global estimates) | Estimated mean salt intake and 95% confidence interval | Ratio between machine learning predicted and global estimates |
|---|---|---|---|---|---|
| Algeria | 2017 | 8.4 (8.4–8.5) | 2010 | 10.7 (9–12.5) | 0.8 |
| Bahamas | 2012 | 9.1 (8.9–9.3) | 2010 | 7.5 (6.2–8.8) | 1.2 |
| Barbados | 2007 | 8.7 (8.4–8.9) | 2010 | 8.6 (7.8–9.4) | 1 |
| Benin | 2015 | 8 (7.8–8.2) | 2010 | 7.1 (6.2–8.1) | 1.1 |
| Botswana | 2014 | 8 (7.9–8.1) | 2010 | 6.3 (5.4–7.4) | 1.3 |
| Cabo Verde | 2007 | 8.1 (8–8.2) | 2010 | 8.1 (6.8–9.7) | 1 |
| Cambodia | 2010 | 7.8 (7.8–7.9) | 2010 | 11 (9.3–12.9) | 0.7 |
| Comoros | 2011 | 8.3 (8.2–8.3) | 2010 | 4.2 (3.5–5) | 2 |

*Appendix 1—table 8 Continued on next page*

*Appendix 1—table 8 Continued*

| Country | Year (machine learning predictions) | Machine learning predicted mean salt intake and 95% confidence interval | Year (global estimates) | Estimated mean salt intake and 95% confidence interval | Ratio between machine learning predicted and global estimates |
|---|---|---|---|---|---|
| Ecuador | 2018 | 8.6 (8.6–8.7) | 2010 | 7.6 (6.4–8.9) | 1.1 |
| Eritrea | 2010 | 6.8 (6.8–6.8) | 2010 | 5.9 (5–7) | 1.2 |
| Ethiopia | 2015 | 7.7 (7.7–7.7) | 2010 | 5.7 (4.9–6.7) | 1.4 |
| Fiji | 2011 | 8.7 (8.6–8.8) | 2010 | 7.2 (6–8.5) | 1.2 |
| Gambia | 2010 | 8.1 (8–8.2) | 2010 | 7.7 (6.5–8.9) | 1.1 |
| Grenada | 2011 | 8.5 (8.4–8.6) | 2010 | 6.5 (5.5–7.7) | 1.3 |
| Guyana | 2016 | 8.5 (8.4–8.6) | 2010 | 6.1 (5.1–7.3) | 1.4 |
| Iraq | 2015 | 8.9 (8.8–8.9) | 2010 | 9.4 (8–11.2) | 0.9 |
| Kenya | 2015 | 8 (7.9–8.1) | 2010 | 3.7 (3.4–4) | 2.2 |
| Kiribati | 2016 | 9 (8.9–9.1) | 2010 | 5.6 (4.6–6.7) | 1.6 |
| Kuwait | 2014 | 9 (8.9–9.1) | 2010 | 9.7 (8.7–10.8) | 0.9 |
| Kyrgyzstan | 2013 | 8.6 (8.5–8.6) | 2010 | 13.4 (11.4–15.8) | 0.6 |
| Lao People's Democratic Republic | 2013 | 7.9 (7.8–8) | 2010 | 11.1 (9.4–13.2) | 0.7 |
| Lesotho | 2012 | 8.4 (8.3–8.5) | 2010 | 6.6 (5.5–7.8) | 1.3 |
| Liberia | 2011 | 8.5 (8.4–8.6) | 2010 | 6.7 (5.6–7.9) | 1.3 |
| Libya | 2009 | 8.7 (8.6–8.8) | 2010 | 10.6 (8.9–12.5) | 0.8 |
| Marshall Islands | 2018 | 9 (9–9.1) | 2010 | 6.4 (5.4–7.5) | 1.4 |
| Mozambique | 2005 | 7.8 (7.6–7.9) | 2010 | 5.6 (4.7–6.6) | 1.4 |
| Myanmar | 2014 | 8 (7.9–8) | 2010 | 11.2 (9.4–13.2) | 0.7 |
| Namibia | 2005 | 7.9 (7.6–8.1) | 2010 | 6.6 (5.6–7.7) | 1.2 |
| Niger | 2007 | 7.7 (7.7–7.7) | 2010 | 7.3 (6.2–8.6) | 1.1 |
| Qatar | 2012 | 9 (8.9–9.1) | 2010 | 10.5 (8.3–12.9) | 0.9 |
| Republic of Moldova | 2013 | 8.5 (8.5–8.6) | 2010 | 9.9 (8.3–11.6) | 0.9 |
| Rwanda | 2013 | 7.9 (7.9–8) | 2010 | 4 (3.3–4.9) | 2 |
| Samoa | 2013 | 9.5 (9.4–9.6) | 2010 | 5.2 (4.6–5.8) | 1.8 |
| Sao Tome and Principe | 2008 | 8.1 (8–8.2) | 2010 | 5.9 (4.9–6.9) | 1.4 |
| Sierra Leone | 2009 | 7.9 (7.8–8) | 2010 | 6.3 (5.3–7.3) | 1.3 |
| Sri Lanka | 2015 | 8 (8–8.1) | 2010 | 9.7 (8.2–11.3) | 0.8 |
| Tajikistan | 2017 | 8.5 (8.4–8.6) | 2010 | 13.5 (11.6–15.7) | 0.6 |
| Timor-Leste | 2014 | 8.2 (7.9–8.4) | 2010 | 11.2 (9.3–13.3) | 0.7 |
| Uganda | 2014 | 7.9 (7.9–8) | 2010 | 5.3 (4.4–6.3) | 1.5 |
| United Republic of Tanzania | 2012 | 7.9 (7.9–8) | 2010 | 6.9 (6.1–7.7) | 1.1 |
| Uruguay | 2014 | 8.5 (8.4–8.5) | 2010 | 6.8 (5.8–8) | 1.2 |
| Vanuatu | 2011 | 8.4 (8.3–8.4) | 2010 | 5.6 (4.8–6.6) | 1.5 |
| Vietnam | 2015 | 7.9 (7.8–7.9) | 2010 | 11.5 (9.5–13.7) | 0.7 |

There are 43 countries in this table; that is, countries included in our analysis that were not available in the previous global work were not included in this table (**Powles et al., 2013**).

**Appendix 1—table 9.** Countries included in the analysis by income group according to the World Bank classification.

| Analysis | World region | Country | Year | Income group |
|---|---|---|---|---|
| Model application | Africa | Algeria | 2017 | Upper-middle |
| Model application | Western Pacific | American Samoa | 2004 | Upper-middle |
| Model application | Americas | Bahamas | 2012 | High |
| Model application | Americas | Barbados | 2007 | High |

*Appendix 1—table 9 Continued on next page*

*Appendix 1—table 9 Continued*

| Analysis | World region | Country | Year | Income group |
|---|---|---|---|---|
| Model application | Africa | Benin | 2015 | Lower |
| Model application | Africa | Botswana | 2014 | Upper-middle |
| Model application | Americas | British Virgin Islands | 2009 | No data |
| Model application | Africa | Cabo Verde | 2007 | Lower-middle |
| Model application | Western Pacific | Cambodia | 2010 | Lower |
| Model application | Americas | Cayman Islands | 2012 | High |
| Model application | Africa | Comoros | 2011 | Lower |
| Model application | Western Pacific | Cook Islands | 2015 | No data |
| Model application | Americas | Ecuador | 2018 | Upper-middle |
| Model application | Africa | Eritrea | 2010 | Lower |
| Model application | Africa | Eswatini | 2014 | Lower-middle |
| Model application | Africa | Ethiopia | 2015 | Lower |
| Model application | Western Pacific | Fiji | 2011 | Lower-middle |
| Model application | Western Pacific | French Polynesia | 2010 | High |
| Model application | Africa | Gambia | 2010 | Lower |
| Model application | Americas | Grenada | 2011 | Upper-middle |
| Model application | Americas | Guyana | 2016 | Upper-middle |
| Model application | Eastern Mediterranean | Iraq | 2015 | Upper-middle |
| Model application | Africa | Kenya | 2015 | Lower-middle |
| Model application | Western Pacific | Kiribati | 2016 | Lower-middle |
| Model application | Eastern Mediterranean | Kuwait | 2014 | High |
| Model application | Eastern Mediterranean | Kyrgyzstan | 2013 | Lower-middle |
| Model application | Western Pacific | Lao People's Democratic Republic | 2013 | Lower-middle |
| Model application | Africa | Lesotho | 2012 | Lower-middle |
| Model application | Africa | Liberia | 2011 | Lower |
| Model application | Eastern Mediterranean | Libya | 2009 | Upper-middle |
| Model application | Western Pacific | Marshall Islands | 2018 | Upper-middle |
| Model application | Africa | Mozambique | 2005 | Lower |
| Model application | Southeast Asia | Myanmar | 2014 | Lower-middle |
| Model application | Africa | Namibia | 2005 | Lower-middle |
| Model application | Western Pacific | Nauru | 2016 | Upper-middle |
| Model application | Africa | Niger | 2007 | Lower |
| Model application | Western Pacific | Niue | 2012 | No data |
| Model application | Western Pacific | Palau | 2013 | Upper-middle |
| Model application | Eastern Mediterranean | Qatar | 2012 | High |
| Model application | Europe | Republic of Moldova | 2013 | Lower-middle |
| Model application | Africa | Rwanda | 2013 | Lower |
| Model application | Western Pacific | Samoa | 2013 | Lower-middle |

*Appendix 1—table 9 Continued on next page*

*Appendix 1—table 9 Continued*

| Analysis | World region | Country | Year | Income group |
|---|---|---|---|---|
| Model application | Africa | Sao Tome and Principe | 2008 | Lower-middle |
| Model application | Africa | Sierra Leone | 2009 | Lower |
| Model application | Southeast Asia | Sri Lanka | 2015 | Lower-middle |
| Model application | Europe | Tajikistan | 2017 | Lower |
| Model application | Southeast Asia | Timor-Leste | 2014 | Lower-middle |
| Model application | Africa | Togo | 2011 | Lower |
| Model application | Western Pacific | Tuvalu | 2015 | Upper-middle |
| Model application | Africa | Uganda | 2014 | Lower |
| Model application | Africa | United Republic of Tanzania | 2012 | Lower |
| Model application | Americas | Uruguay | 2014 | High |
| Model application | Western Pacific | Vanuatu | 2011 | Lower-middle |
| Model application | Western Pacific | Vietnam | 2015 | Lower-middle |
| Model derivation | Europe | Armenia | 2016 | Lower-middle |
| Model derivation | Europe | Azerbaijan | 2017 | Upper-middle |
| Model derivation | Southeast Asia | Bangladesh | 2018 | Lower-middle |
| Model derivation | Europe | Belarus | 2017 | Upper-middle |
| Model derivation | Southeast Asia | Bhutan | 2014 | Lower-middle |
| Model derivation | Southeast Asia | Bhutan | 2019 | Lower-middle |
| Model derivation | Western Pacific | Brunei Darussalam | 2016 | High |
| Model derivation | Americas | Chile | 2017 | High |
| Model derivation | Eastern Mediterranean | Jordan | 2019 | Upper-middle |
| Model derivation | Eastern Mediterranean | Lebanon | 2017 | Upper-middle |
| Model derivation | Africa | Malawi | 2017 | Lower |
| Model derivation | Western Pacific | Mongolia | 2013 | Lower-middle |
| Model derivation | Western Pacific | Mongolia | 2019 | Lower-middle |
| Model derivation | Eastern Mediterranean | Morocco | 2017 | Lower-middle |
| Model derivation | Southeast Asia | Nepal | 2019 | Lower-middle |
| Model derivation | Western Pacific | Solomon Islands | 2015 | Lower-middle |
| Model derivation | Eastern Mediterranean | Sudan | 2016 | Lower-middle |
| Model derivation | Western Pacific | Tokelau | 2014 | No data |
| Model derivation | Western Pacific | Tonga | 2017 | Upper-middle |
| Model derivation | Europe | Turkmenistan | 2018 | Upper-middle |
| Model derivation | Africa | Zambia | 2017 | Lower-middle |

Source: World Bank (https://datahelpdesk.worldbank.org/knowledgebase/articles/906519-world-bank-country-and-lending-groups).

Sample = 103,462 observations

Sample = 101,938 observations
[age between 15-69 years]

Sample = 58,846 observations
[complete-case in height, weight,
blood pressure, urine creatinine and
sodium]

Sample = 58,820 observations
[plausible values in height, weight,
blood pressure and BMI]

Sample = 58,815 observations
[non- pregnant women]

Sample = 56,578 observations
[excluded Georgia and Afghanistan
because inconsistent urine creatinine
or missing data]

Sample = 50,940 observations
[plausible urine creatinine values]

Sample = 50,934 observations
[only positive values on estimated salt
intake]

Sample = 49,776 observations
[estimated salt intake values within 3
standard deviations from their mean]

Total sample = 49,776 observations
[48.1% of initial sample size]

**Appendix 1—figure 1.** Flowchart of data cleaning and inclusion criteria for model derivation.

## Appendix 2

### Expanded methods

#### Characteristics of the surveys included in the analysis

We analyzed WHO STEPS surveys and one national health survey (Chile) (*World Health Organization, 2021b*; *Wang et al., 2020*). These surveys included a random sample of the general population and can deliver nationally representative estimates. These are household surveys that stratify by the first administrative level in the country (e.g., region); within this level, further stratification may occur by, for example, urban/rural location. Then, a random sample of census tracts, villages, neighborhoods, or other similar division is selected. In each of these primary sampling units, households are randomly sampled for the interview.

All surveys followed standard procedures (*World Health Organization, 2021b*; *Wang et al., 2020*). Briefly, participants were given a small container along with instructions for the urine collection; the next day, participants brought the urine sample to a designated place. Then, urine samples were analyzed at a laboratory by a trained technician.

#### Overview

We worked with a structured dataset that mostly had numeric attributes (variables). Given our study problem, we opted for a supervised learning model because there was a target attribute (i.e., salt consumption at the subject level); specifically, we conducted a supervised regression because the target attribute was a numeric variable. For the machine learning analyses, we used Python and the Scikit-Learn library.

First, we developed a pipeline for data management and model development. This way, we followed a consistent and transparent methodology to secure an optimal model for the training set and that would adequately generalize to other (unseen) datasets. *Appendix 2—figure 1* depicts the pipeline we developed: (i) we studied the available data and where needed, we did a one-hot encoding; (ii) we did feature importance analysis; (iii) we chose and tried different scaling and transformation methods, so that all variables would be in the same scale or units; (iv) we tried a set of machine learning models, including a customized neural network; and (v) we forecasted (predicted) the attribute of interest (salt consumption at the subject level) in an unseen dataset (i.e., not used for model training). Notably, we went backward and forward (see arrows in the figure) between the four first stages until we reached the best combinations and results for each model. In the following sections, we will describe each of these five stages.

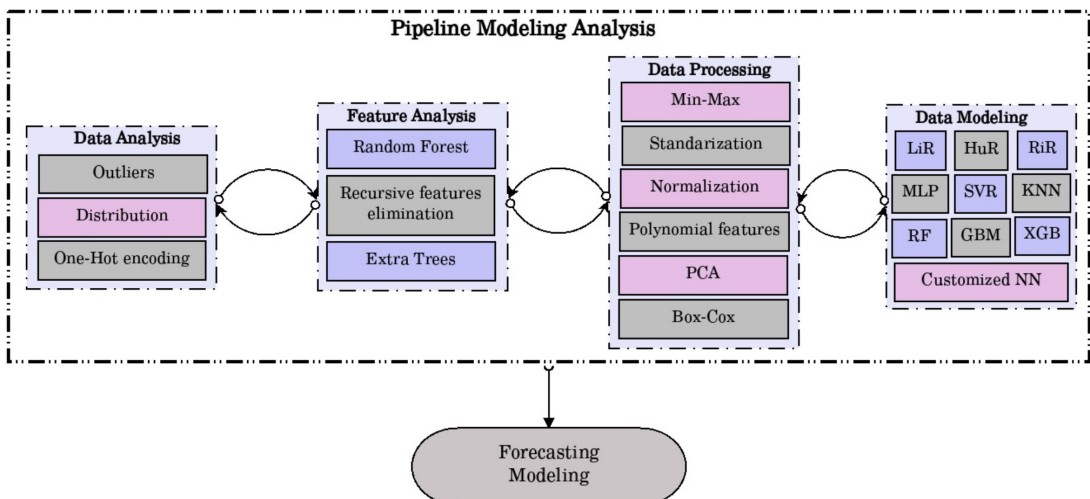

**Appendix 2—figure 1.** Pipeline for data management and model development. PCA, primary component analysis; LiR, linear regression; HuR, Hubber regressor; RiR, ridge regressor; MLP, multilayer perceptron; SVR, support vector regressor; KNN, *k*-nearest neighbors; RF, random forest; GBM, gradient boost machine; XGB, extreme gradient boosting; NN, neural network.

## Data analysis

This was an exploratory analysis to understand the dataset and its characteristics. We worked with a complete-case dataset; in other words, we excluded missing observations in the variables considered in the analysis. Consequently, we did not do any data imputation analysis.

We explored the distribution of all numerical variables, which were in different units and scales; this exploratory analysis informed the choices of data processing methods (e.g., Box-Cox) implemented in the third stage.

## Feature importance analysis

Even though we followed expert knowledge to select a reduced, though relevant number of predictors to be included in the regression model, we conducted feature importance analyses to understand the role each predictor would play in the model. This process aimed to eliminate variables that would not carry substantial information for the model. We used random forest, recursive feature elimination, and extra trees. Consistently, these three methods suggested that all the chosen predictors would contribute to a better model.

## Data processing

As described in the data analysis section (first stage), numeric variables were in different units and scales; therefore, these variables needed to be scaled or transformed. This scaling would also help to find a better prediction model. It is common knowledge that machine learning models would perform differently (and better) depending on data transformation methods. We did (i) min-max whereby numeric variables were scaled to a range between 0 and 1; (ii) standardization; (iii) normalization: (iv) polynomial features of degree 2 (quadratic polynomial); (v) principal component analysis with three components and explained variance of ≥0.95; and (vi) Box-Cox.

## Data modeling

There are several machine learning algorithms for a supervised regression model. Those that we used, and that are depicted in *Appendix 1—figure 1*, yielded much better results and were studied in detail. That is, at the beginning of our work we explored other algorithms, though these did not perform well and were not considered thereafter. The algorithms we considered were (i) linear regression (LiR); (ii) Hubber regressor (HuR); (iii) ridge regressor (RiR); (iv) multilayer perceptron (MLP); (v) support vector regressor (SVR); (vi) *k*-nearest neighbors (KNN); (vii) random forest (RF); (viii) gradient boost machine (GBM); and (ix) extreme gradient boosting (XBG).

In addition to these nine machine learning algorithms, we also implemented a neural network (see *Appendix 2—figure 2*). This neural network was optimized empirically. We used a batch size = 256; epochs = 300; and optimizer = 'adam.' The neural network was implemented in Python using the Keras library.

For each model and processing method (see 'Data processing' section), we studied the $R^2$, mean absolute error (MAE), and root mean square error (RMSE). As shown in *Appendix 1—table 1*, all algorithms showed a similar performance. Because all the algorithms had an equivalent performance, the chosen one needed to be defined at the forecasting stage; that is, the one that would generalize better to new (unseen) data.

## Forecasting modeling

This stage implies studying the predicted results in new (unseen) data (i.e., data not used for model training). For this stage, we used the validation and test datasets. We chose the model that yielded predictions closest to the observed results. In this line, we compared the mean difference between the observed and predicted mean salt intake results (i.e., observed – predicted) across all prediction algorithms.

We observed there was no unique algorithm that had the mean difference closest to zero in men and women at the same time (*Appendix 1—table 2*). The HuR algorithm had the mean difference closest to zero in both sexes combined (mean difference = –0.0019), the RiR algorithm performed the best in men (mean difference = 0.0063), and in women the HuR algorithm showed the best results (mean difference = 0.0082).

To support our decision process, we plotted the mean differences in men and women for each survey (*Appendix 2—figure 3*); this figure only included the predictions based on the top three

algorithms (HuR, MLP, and customized NN). We counted how many times (i.e., number of surveys) each algorithm had the mean difference closest to zero.

Because the HuR algorithm had the mean difference closest to zero in both sexes combined and it was among the top five algorithms in men and women (*Appendix 1—table 2*), we decided to choose the HuR algorithm. Additionally, predictions based on the HuR algorithm were the closest to zero across surveys (*Appendix 2—figure 3*). These analyses were performed in R (version 4.0.3).

## Algorithm application

To make the predictions in the new 54 datasets without information about urine samples, we used the HuR model (i.e., ML algorithm and predictors) developed following the methods above described (see 'Forecasting modeling' section). We re-trained the model with the full dataset used for model development and validation (i.e., train, validated, and test dataset pooled), and then predicted the outcome (i.e., mean salt intake) in the 54 new datasets.



**Appendix 2—figure 2.** Neural network implementation.

**Appendix 2—table 1.** Performance of each algorithm and processing method.

| Algorithm | Processing | $R^2$ | MAE | RMSE |
|---|---|---|---|---|
| LiR | Polynomial (g = 2) | 0.447 | 1.1138 | 1.4451 |
| HuR | Standardized | 0.447 | 1.1132 | 1.4442 |
| RiR | Polynomial (g = 2) | 0.446 | 1.1147 | 1.4459 |
| MLP | Min-max | 0.451 | 1.1101 | 1.4395 |
| SVR | Min-max | 0.446 | 1.0988 | 1.4459 |
| KNN | Standardized | 0.421 | 1.1426 | 1.4779 |
| RF | Polynomial (g = 2) | 0.417 | 1.1474 | 1.4835 |
| GBM | Min-max | 0.447 | 1.1147 | 1.4447 |
| XGB | Min-max | 0.431 | 1.1293 | 1.4646 |
| Customized NN | Box-Cox | 0.461 | 1.0953 | 1.4156 |

MAE: mean absolute error. RMSE: root mean square error. LiR: linear regression. HuR: Hubber regressor. RiR: ridge regressor. MLP: multilayer perceptron. SVR: support vector regressor. KNN: *k*-nearest neighbors. GBM: gradient boost machine. XGB: extreme gradient boosting. NN: neural network; RF: random forest.

**Appendix 2—table 2.** Mean difference between observed and predicted salt intake by sex across all machine learning algorithms.

| Machine learning algorithm | Mean difference between observed and predicted mean salt intake | Sex |
|---|---|---|
| CNN_boxcox | –0.0109 | Both sexes |
| CNN_standardize | –0.0075 | Both sexes |
| GBR_boxcox | 0.1373 | Both sexes |
| GBR_minmax | 0.1198 | Both sexes |
| GBR_orig | –0.0252 | Both sexes |

*Appendix 2—table 2 Continued on next page*

*Appendix 2—table 2 Continued*

| Machine learning algorithm | Mean difference between observed and predicted mean salt intake | Sex |
|---|---|---|
| GBR_standardized | 0.1231 | Both sexes |
| HuR_boxcox | 0.0389 | Both sexes |
| HuR_standardized | –0.0019 | Both sexes |
| KNN_boxcox | 0.0144 | Both sexes |
| KNN_standardized | –0.0172 | Both sexes |
| LiR_poly | –0.0292 | Both sexes |
| MLP_boxcox | –0.0069 | Both sexes |
| MLP_minmax | –0.019 | Both sexes |
| MLP_standardized | –0.0174 | Both sexes |
| RF_poly | –0.0479 | Both sexes |
| RiR_poly | –0.0304 | Both sexes |
| SVR_minmax | 0.1137 | Both sexes |
| XGB_boxcox | 0.0389 | Both sexes |
| XGB_orig | –0.0312 | Both sexes |
| XGB_standardized | –0.0329 | Both sexes |
| CNN_boxcox | 0.088 | Men |
| CNN_standardize | 0.0699 | Men |
| GBR_boxcox | 0.1591 | Men |
| GBR_minmax | 0.1381 | Men |
| GBR_orig | 0.0197 | Men |
| GBR_standardized | 0.1444 | Men |
| HuR_boxcox | 0.0265 | Men |
| HuR_standardized | –0.0119 | Men |
| KNN_boxcox | 0.0612 | Men |
| KNN_standardized | 0.0179 | Men |
| LiR_poly | 0.0069 | Men |
| MLP_boxcox | 0.0512 | Men |
| MLP_minmax | –0.0104 | Men |
| MLP_standardized | –0.0249 | Men |
| RF_poly | –0.0129 | Men |
| RiR_poly | 0.0063 | Men |
| SVR_minmax | 0.1265 | Men |
| XGB_boxcox | 0.0265 | Men |
| XGB_orig | 0.0147 | Men |
| XGB_standardized | 0.0069 | Men |
| CNN_boxcox | –0.1097 | Women |

*Appendix 2—table 2 Continued on next page*

*Appendix 2—table 2 Continued*

| Machine learning algorithm | Mean difference between observed and predicted mean salt intake | Sex |
|---|---|---|
| CNN_standardized | –0.085 | Women |
| GBR_boxcox | 0.1155 | Women |
| GBR_minmax | 0.1015 | Women |
| GBR_orig | –0.07 | Women |
| GBR_standardized | 0.1018 | Women |
| HuR_boxcox | 0.0514 | Women |
| HuR_standardized | 0.0082 | Women |
| KNN_boxcox | –0.0324 | Women |
| KNN_standardized | –0.0524 | Women |
| LiR_poly | –0.0653 | Women |
| MLP_boxcox | –0.0649 | Women |
| MLP_minmax | –0.0276 | Women |
| MLP_standardized | –0.0098 | Women |
| RF_poly | –0.0828 | Women |
| RiR_poly | –0.0671 | Women |
| SVR_minmax | 0.101 | Women |
| XGB_boxcox | 0.0514 | Women |
| XGB_orig | –0.0771 | Women |
| XGB_standardized | –0.0727 | Women |

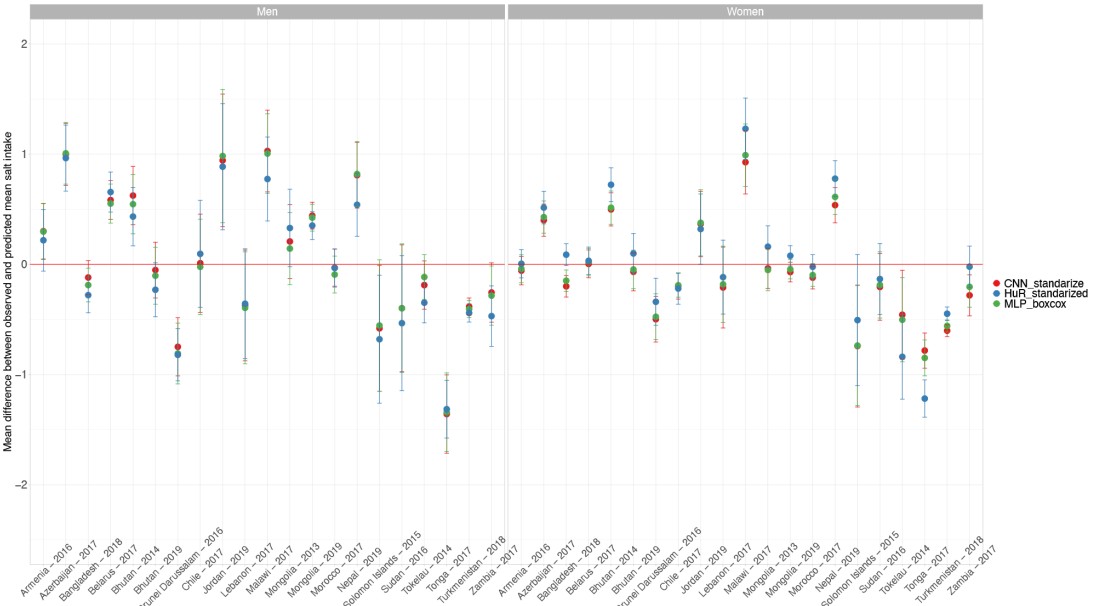

**Appendix 2—figure 3.** Comparison between mean difference between observed and predicted salt intake across the best algorithms. CNN, customized neural network; HuR: Hubber regressor; MLP, multilayer perceptron.

