## [Editor Report]

Salt intake is a major determinant of volume status, blood pressure values, and congestion, but its estimation is challenging because of the need of measuring 24-h urinary sodium excretion over a number of days, which is unfeasible in most countries. The demonstration of the feasibility of estimating accurately salt intake at the population level using artificial intelligence starting from simple and widely available variable is therefore important for epidemiological and intervention studies in which salt intake is a major player, particularly, but not only, in countries experiencing economic hardships.

---

## [Decision Letter]

**Decision letter after peer review:**

Thank you for submitting your article "Estimating salt consumption in 49 low-and middle-income countries: Development, validation and application of a machine learning model" for consideration by *eLife*. Your article has been reviewed by 2 peer reviewers, including Gian Paolo Rossi as Reviewing Editor and Reviewer #1, and the evaluation has been overseen by a Senior Editor.

Both reviewers have found your manuscript of interest and potential relevance, as estimation of salt consumption by artificial intelligence in the population is a general problem. The Reviewers also agreed that this is more important for the low-mid income countries, which cannot afford measurements of sodium in a 24 hour urine collection or in a spot urinary sample. However, since the problem of estimating salt intake is not confined to such countries, a valuable addition that can increase the scientific merit of the study would be to provide data also on middle and high income countries.

They regarded as strengths of the study the development of a tool for estimating the sodium intake applicable to each country, particularly to those where it is difficult to collect urine specimens, and also the novel machine learning approach applied to 19 WHO STEPS surveys including more than 45,000 people.

However, some methodological limitations were also noted, including the fact that your ML model was developed and validated using, as reference, data obtained from a 'golden' (spot urine samples), not a gold standard method (i.e. 24-hour urine sample). One Reviewer underlined that all equations, including the Intersalt's used as outcome in this study, can imply a bias in predicting 24h U-Na^+^ excretion, even with use of correction formulas (see Charlton KE, Schutte A et al., 2020). Hence, the finding that mean salt consumption predicted by the supervised ML model did not differ significantly from the mean observed value, should be validated with the Na^+^ intake determined by the 24h U-Na^+^ excretion.

*Reviewer #1 (Recommendations for the authors):*

I find these results to be important. The manuscript is well written and the methodology seems to be correct. However, since the problem of estimating salt intake is not confined to low income countries, a valuable addition that can increase the scientific merit of the study would be to provide data also on middle and high income countries.

*Reviewer #2 (Recommendations for the authors):*

In this study Guzman-Vilca et al., investigated if a machine learning (ML) model based on predictors that are routinely available in large scale surveys could predict salt intake in low- and middle-income countries (LMICs), and could be an appropriate tool to estimate sodium/salt intake in the national health surveys.

This is an interesting study that moves from the need of estimating sodium intake in countries that have no access to urine samples, and exploits a novel method to pursue the aim. However, the study suffers a major methodological limitation that should be deeply considered.

Major criticisms:

The Authors trained, tested and validated the ML model using data obtained from ‘golden standard’ methods (spot urine samples), not a gold standard method as reference (i.e. 24-hour urine sample) as recommended by STARD (Bossuyt PM. Ann Intern Med 2003). Even though not updated, there are survey from LMICs considering 24h U-Na^+^ excretion. This is a methodological limitation that should be amended.

Moreover, all equations, including the Intersalt used as outcome in this study, can implies a bias in predicting 24h U-Na^+^ excretion, even after using correction formulas (e.g. see Charlton KE, Schutte A et al., 2020). Hence, the mean salt consumption predicted by the supervised ML model, which was found not significantly different from the mean observed value, could be significantly different from the intake calculated with 24h U-Na^+^ excretion. Finally, the deviations from WHO recommendations (<5g daily) in the LMICs could be different from those resulting from the golden (not gold) standard approach-ML based model.

The quality of each survey, including the 19 surveys used for ML training and validation and those 49 used for estimating salt intake, should be preliminary evaluated using a validated scoring system before data processing as QUADAS-2. Quality could be crucial to understand differences between observed and predicted values.

Only a sub-analysis by sex was reported. It could be interesting to also evaluate how the ML model works at different ages, BP and BMI values.

Age range 15-69. Data from old/very old people were not considered at all. It is unclear if the exclusion of old people is related to the STEPS templates that include limited classes of age, or not.

Figure Pipeline Modeling Analysis. Please introduce abbreviations and provide a brief legend.

Last Figure (number is missing). Please add number, title and legend; enlarge dots and text on the right.

Tables. Please remove decimals from SBP and DBP.

---

## [Author Response]

Both reviewers have found you manuscript of interest and potential relevance, as estimation of salt consumption by artificial intelligence in the population is a general problem. The Reviewers also agreed that this is more important for the low-mid income countries, which cannot afford measurements of sodium in a 24 hour urine collection or in a spot urinary sample. However, since the problem of estimating salt intake is not confined to such countries, a valuable addition that can increase the scientific merit of the study would be to provide data also on middle and high income countries.

We are glad the editors and reviewers found merit in our work. We agree that this is a relevant topic. We also agree that a work including many more countries, regardless of income group and world region, would be of higher scientific value.

In this revised version we analysed more surveys, including high-income countries. This table shows the income group of each country, and whether it was included in the derivation or application phase of our analysis. Consequently, throughout our manuscript where we referred to the analysed surveys, we changed “LMICs” to “countries” (or “surveys”).

This table is now included in the Appendix 1 – table 9 and described in the text:

“According to the World Bank classification (Appendix 1 – table 9), there were nine high-income countries (two in model derivation and seven in model application); 16 low-income countries (one in model derivation and 15 in model application); 26 lower-middle income countries (9 in model derivation and 17 in model application); and 18 upper-middle income countries (six in model derivation and 12 model application). There were four countries (one in model derivation and three in model application) without income classification (British Virgin Islands, Cook Islands, Niue and Tokelau).” [pp. 11-12]

The title of the manuscript was updated as well:

“Development, validation and application of a machine learning model to estimate salt consumption in 54 countries” [p. 01]

Throughout the manuscript we updated the total number of surveys included in the derivation or application phase.

In conclusion, in this revised submission we increased the number of analysed surveys (see Author response table 1). Overall, for model development we analysed two additional surveys (1 upper-middle and 1 high income). For model application, we analysed five additional surveys (2 low, 1 lower-middle, 1 upper-middle and 1 high income). That is, of the seven new surveys included in the revised analysis, four were upper-middle or high-income countries.

**Author response table 1. sa2table1:** 

	Income						
	Total	Low	Lower-middle	Upper-middle	High	No group	
Model development	Original	17	1	9	5	1	1
	Revision	19	1	9	6	2	1
	Difference	+2			+1	+1	
Model application	Original	49	13	16	11	6	3
	Revision	54	15	17	12	7	3
	Difference	+5	+2	+1	+1	+1	

Furthermore, of all upper-middle-income countries in the world, 32% (18/56) were included in the analysis; of all high-income countries in the world, 11% (9/83) were included in the analysis; of all lower-middle-income countries in the world, 52% (26/50) were included in the analysis; of all low-income-countries in the world, 55% (29) were included in the analysis. Although still not covering all countries in the world, our work includes countries in all income groups as suggested by the reviewers. We appreciate the editor’s and reviewers’ suggestions which improved our work.

Reviewer #1 (Recommendations for the authors): I find these results to be important. The manuscript is well written and the methodology seems to be correct. However, since the problem of estimating salt intake is not confined to low income countries, a valuable addition that can increase the scientific merit of the study would be to provide data also on middle and high income countries.

We understood the reviewer’s concern and appreciate his suggestion. To meet his expectations, we conducted additional work. We trust the reviewers and editors will find this additional work satisfactory. We are happy to receive further feedback as needed.

In this revised version we analysed more surveys, including more high-income countries. Please, refer to the tables and text in our response to the editor’s comment.

For our analysis we targeted population-based nationally representative surveys; in addition, these surveys must be available for independent analysis free of charge and without significant burden to access (e.g., data sharing agreements that may involve institutional signatures). This was further detailed in the manuscript:

“We sought surveys that met these two criteria: (i) nationally representative health surveys (i.e., community or sub-national surveys were not included); and (ii) surveys that were open access or that could be accessed without significant administrative burden (e.g., data sharing agreements that may involve institutional signatures).” [p. 11]

The WHO STEPS surveys met these criteria. In response to the reviewer’s comment we sought additional surveys following these procedures.

A work by the Global Burden of Diseases Nutrition and Chronic Diseases Expert Group delivered estimates of 24-hour urinary sodium excretion for 187 countries. We reviewed the data sources included in their analysis. These data sources were not nationally representative or were not accessible for independent re-analyses. Therefore, we could not incorporate any of these data sources in our work.

In a systematic review by Sudhir Raj Thout and colleagues, they sought nationally representative surveys with 24-hour urine collection. While almost all data came from high-income countries, these data were not open access for independent re-analyses. Therefore, we could not incorporate any of these data sources in our work.

In a systematic review by our research group, which focused on population-based surveys from Latin America regardless of the method of urine collection (24-hour vs otherwise), there were three national health surveys. We accessed data from a national health survey in Chile (high-income country). Therefore, we included this survey in our analysis for model derivation.

In a systematic review by Oyinlola Oyebode and collagues, they summarised studies from Africa with urine and sodium information. We checked the references they found though none were available for independent re-analyses. Therefore, we could not incorporate any of these data sources in our work.

This process was explained in the Materials and methods section.

“To identify additional data sources we searched the original publications included in one global analysis and three systematic reviews about sodium/salt consumption at the population level. This search led to the identification of the national health survey included in the model derivation. All other data sources included in those references did not meet our selection criteria.

In conclusion, our ML model was developed based on 21 surveys (20 WHO STEPS and 1 national health survey). Then, our ML model was applied to 54 WHO STEPS survey to compute the mean daily salt consumption at the population level.” [p. 11]

Reviewer #2 (Recommendations for the authors):In this study Guzman-Vilca et al., investigated if a machine learning (ML) model based on predictors that are routinely available in large scale surveys could predict salt intake in low- and middle-income countries (LMICs), and could be an appropriate tool to estimate sodium/salt intake in the national health surveys.This is an interesting study that moves from the need of estimating sodium intake in countries that have no access to urine samples, and exploits a novel method to pursue the aim. However, the study suffers a major methodological limitation that should be deeply considered.Major criticisms:The Authors trained, tested and validated the ML model using data obtained from 'golden standard' methods (spot urine samples), not a gold standard method as reference (i.e. 24-hour urine sample) as recommended by STARD (Bossuyt PM. Ann Intern Med 2003). Even though not updated, there are survey from LMICs considering 24h U-Na^+^ excretion. This is a methodological limitation that should be amended.

For our analyses we targeted surveys that met these two criteria: (i) nationally representative health surveys; and (ii) surveys that were in the public domain (open access) or that could be accessed without significant administrative burden (e.g., complex data sharing agreements that may involve institutional signatures). This was further detailed:

“We sought surveys that met these two criteria: (i) nationally representative health surveys (i.e., community or sub-national surveys were not included); and (ii) surveys that were open access or that could be accessed without significant administrative burden (e.g., data sharing agreements that may involve institutional signatures).” [p. 11]

Despite our experience working in data pooling projects,[4] and the additional work to identify more relevant surveys, we are not aware of any nationally representative health survey with 24hour urine specimens that are in the public domain available for independent re-analyses. In other words, there are no surveys with 24-hour urine samples that would meet our two inclusion criteria.

We reviewed the references of one global modelling study and three recent systematic reviews. We carefully checked the data sources they summarised including those with 24hour urine samples, and we could not find any data sources that were available for independent re-analyses (open access). In conclusion, even though we wanted to include surveys with 24hour urine samples, and we exhaustively sought these surveys, there are no nationally representative surveys with 24-hour urine samples available for independent re-analyses.

The fact that we only analysed spot urine samples is a limitation which was acknowledged in our original submission. We have further highlighted this limitation throughout our work.

“As an alternative, equations have been developed to estimate sodium/salt intake based on spot urine (SU) samples. *Although these equations may not deliver identical results to those based on 24-hour urine samples at the individual level,* at the population level the difference between SU samples and 24-hour samples appears to be small.” [p. 03]

“*The fact that we analysed SU samples is a limitation of our work and the results should be interpreted accordingly.”* [p. 12]

“Of note, our outcome variable was informed by SU samples and not by 24-hour urine samples (gold-standard to assess salt consumption). *Results should be interpreted according to this limitation.*” [p. 13]

“It should be noted that we analysed SU samples. These are not the gold standard to assess salt consumption. *Results should be interpreted in light of this limitation,* considering that our model aimed to deliver estimates at the population level (not individual level).” [p. 07]

“First and foremost, *urine data was based on a spot sample, which is not the gold standard (24hour urine sample) to measure daily salt consumption*.” [pp. 09-10]

We trust the editors and reviewers will still find value in our work. We acknowledged this is a limitation of our work, which was transparently highlighted throughout the manuscript. Unfortunately, there is nothing we can do to overcome this limitation due to the lack of open-access data with 24-hour urine samples. This is a shared limitation with a previous global analysis in which they had to combine 24-hour urine samples and diet assessment tools because of the global lack of data on 24-hour urine samples: “…assessment methods included 24 h urinary excretion measurements, a diet assessment tool (e.g., diet record, diet recall, food frequency questionnaire…”.^10^)

The fact that we only analysed spot urine samples is a limitation. This was acknowledged throughout our manuscript. This is also an observation which calls to produce nationally representative health surveys with 24-hour urine samples, and make them available for independent re-analyses. This argument was further elaborated in:

“While this –re-analysis of SU sample rather than 24-hour urine samples– is a limitation of our work, it is also an observation showing the lack of nationally representative surveys with 24hour urine samples available for independent re-analyses.” [p. 10]

Moreover, all equations, including the Intersalt used as outcome in this study, can implies a bias in predicting 24h U-Na^+^ excretion, even after using correction formulas (e.g. see Charlton KE, Schutte A et al., 2020). Hence, the mean salt consumption predicted by the supervised ML model, which was found not significantly different from the mean observed value, could be significantly different from the intake calculated with 24h U-Na^+^ excretion. Finally, the deviations from WHO recommendations (<5g daily) in the LMICs could be different from those resulting from the golden (not gold) standard approach-ML based model.

Despite our efforts to systematically identify nationally representative health surveys that are in the public domain and included 24-hour urine samples, we did not find any. Please, refer to our previous answer for further details on this subject.

We agree with the observation that the equations can imply bias in estimating the 24-hour salt consumption. This was specifically acknowledged in a new sentence in the introduction.

“As an alternative, equations have been developed to estimate sodium/salt intake based on spot urine (SU) samples. *Although these equations may not deliver identical results to those based on 24-hour urine samples at the individual level,* at the population level the difference between SU samples and 24-hour samples appears to be small.” [p. 03]

However, we would like to rise the following argument. Our focus was the population, not the individual. We aimed to provide a machine learning model which learnt from individual-level data to provide mean estimates at the population level (i.e., overall mean in the country). We did not produce a machine learning model to be used to predict salt consumption for each individual.

The reason why we did not develop a model to be used in individuals relies on the remarks made by the reviewer: potential disagreements between spot urine samples and the gold standard (24-hour urine samples). However, at the population level, the mean appears to be similar between spot urine samples and 24-hour urine samples.[6,7] For example, “Overall average population salt intake estimated from 24-h urine samples was 9.3 g/day compared with 9.0 g/day estimated from the spot urine samples.”^13^

Using spot urine samples is indeed a limitation, which was fully acknowledged throughout the manuscript (please, refer to our previous answer). However, at the population level, the differences between spot urine samples and 24-hour urine samples may be small. We have included these arguments in the manuscript.

“As an alternative, equations have been developed to estimate sodium/salt intake based on spot urine (SU) samples. Although these equations may not deliver identical results to those based on 24-hour urine samples at the individual level, at the population level the difference between SU samples and 24-hour samples appears to be small.” [p. 03]

“If we could (accurately) estimate sodium/salt intake at the population level based on variables that are routinely available in national health surveys (e.g., weight or blood pressure), mean sodium/salt intake at the population level in countries that currently lack urine data (i.e., 24-hour or spot) could be computed using these available predictors. Advanced analytic techniques like machine learning (ML) could make accurate predictions, and inform about the mean sodium/salt intake at the population level. We developed a ML predictive model to estimate mean salt intake at the population level (not at the individual level) using routinely available variables in national health surveys” [p. 04]

“If the model were indeed accurate, then it could be applied to national surveys without urine samples but with the relevant predictors to inform about the mean salt consumption in the overall population.” [p. 12]

“However, we aimed to develop a model that can be used to predict mean estimates at the population level, not at the individual level. In other words, our model should not be applied to a patient to estimate his/her salt consumption. Our model should be applied to survey data to compute the mean sodium/salt consumption in the population (not in individuals). Empirical evidence suggests that, at the population level, mean estimates based on SU samples and on 24-hour urine samples are similar.” [p. 12]

“While SU samples may not be the best approach to estimate salt consumption at the individual level, at the population level the means estimated based on SU samples and 24-hour urine samples are similar. Therefore, the limitation of using SU samples only may have had little impact on our mean estimates which are the country level, not at the individual level.” [p. 10]

“We developed a machine learning (ML) model to predict salt consumption at the population level based on simple predictors…We applied the ML model to 54 new surveys to quantify the mean salt consumption in the population.” [abstract]

The quality of each survey, including the 19 surveys used for ML training and validation and those 49 used for estimating salt intake, should be preliminary evaluated using a validated scoring system before data processing as QUADAS-2. Quality could be crucial to understand differences between observed and predicted values.

We apologise unreservedly, but we are not sure how to address this comment. We tried our best, and we are happy to receive further guidance from the reviewer and editor if needed.

To guide our response, we reviewed publications in which they analysed multiple WHO STEPS surveys and other nationally representative surveys. Author response table 2 shows recent publications using multiple national health surveys where we did not find any similar preliminary evaluation.

**Author response table 2. sa2table2:** 

Teufel F, et al. Body-mass index and diabetes risk in 57 low-income and middle-income countries: a cross-sectional study of nationally representative, individual-level data in 685 616 adults. Lancet. 2021;398(10296):238–48.
Flood D, et al., The state of diabetes treatment coverage in 55 low-income and middle-income countries: a cross-sectional study of nationally representative, individual-level data in 680 102 adults. The L,ancet Healthy Longevity. 2021;2(6):e340–51.
Peiris D, et al., Cardiovascular disease risk profile and management practices in 45 lowincome and middle-income countries: A cross-sectional study of nationally representative individual-level survey data. PLoS Med. 2021;18(3):e1003485.
Davies JI, et al., Association between country preparedness indicators and quality clinical care for cardiovascular disease risk factors in 44 lower- and middle-income countries: A multicountry analysis of survey data. PLoS Med. 2020;17(11):e1003268.
Seiglie JA, et al., Diabetes Prevalence and Its Relationship With Education, Wealth, and BMI in 29 Low- and Middle-Income Countries. Diabetes Care. 2020;43(4):767–75.
Geldsetzer P, et al., The state of hypertension care in 44 low-income and middle-income countries: a cross-sectional study of nationally representative individual-level data from 1·1 million adults. Lancet. 2019;394(10199):652–62.
Manne-Goehler J, et al., Health system performance for people with diabetes in 28 low- and middle-income countries: A cross-sectional study of nationally representative surveys. PLoS Med. 2019;16(3):e1002751.

Nonetheless, in comparison to our original submission, these publications included further details about the sampling design and data collection of the main variable(s). We included the following text in the Appendix 2. These lines summarized the sampling design and data collection processes of the surveys we pooled. References to the manuals and further documentation about these surveys are provided as well.

“We analysed WHO STEPS surveys and one national health survey (Chile). These surveys included a *random sample* of the general population and can deliver nationally representative estimates. These are household surveys that *stratify* by the first administrative level in the country (e.g., region); within this level, further stratification may occur by, for example, urban/rural location. *Then, a random sample of census tracts, villages, neighbourhoods or other similar division, is selected.* In each of these primary sampling units, *households are randomly* sampled for the interview.

All surveys followed standard procedures. Briefly, *participants were given a small container along with instructions for the urine collection*; the next day, participants brought the urine sample to a designated place. Then, urine samples were analysed at a laboratory by a trained technician.”

We are also doubtful about the use of QUADAS-2 tool. This tool is for diagnostic accuracy studies. As we discussed in our previous response, we did not develop a diagnostic tool (i.e., a machine learning model for the individuals). Rather, we aimed to develop a machine learning algorithm that could leverage on survey data to compute the mean at the population level. We included this idea in the manuscript:

“We *did not develop a diagnostic tool* to replace SU or 24-hour urine samples.” [p. 12]

Other criticisms and suggestions"Only a sub-analysis by sex was reported. It could be interesting to also evaluate how the ML model works at different ages, BP and BMI values.

We included these results. The following table is now available in the Appendix 1 – Table 3 and referenced in the Results in the main document.

“The mean observed salt intake was higher in people aged ≥30 years (7.9 g/day vs 8.4 g/day, p<0.05 for independent T-test), and so was for people with raised blood pressure (≥140/90 mmHg) (8.7 g/day vs 8.2 g/day, p<0.05). The mean salt consumption was also different across BMI categories (p<0.05 for ANOVA test). The same profile was found for predicted mean salt intake (Appendix 1 – Table 3).” [p. 05]

Age range 15-69. Data from old/very old people were not considered at all. It is unclear if the exclusion of old people is related to the STEPS templates that include limited classes of age, or not.

Yes, it is because of the age structure of the surveys we re-analysed. This was clarified.

“Our complete-case analysis was restricted to men and non-pregnant women aged between *15 and 69 years because of data availability*.” [p. 13]

Figure Pipeline Modeling Analysis. Please introduce abbreviations and provide a brief legend.

The abbreviations were spelled out.

“PCA: primary component analysis; LiR: Linear Regression; HuR: Hubber Regressor; RiR: Ridge Regressor; MLP: Multilayer Perceptron; SVR: Support Vector Regressor; KNN: kNearest Neighbors; RF: Random Forest; GBM: Gradient Boost Machine; XGB: Extreme Gradient Boosting; NN: Neural Network.” [Appendix 2 – Figure 1]

Last Figure (number is missing). Please add number, title and legend; enlarge dots and text on the right.

Appendix 2 – Figure 3 has been modified as suggested.

Tables. Please remove decimals from SBP and DBP.

In all Appendix 1 – tables, SBP and DBP are presented without decimal places.

References

Powles J, Fahimi S, Micha R, Khatibzadeh S, Shi P, Ezzati M, et al., Global, regional and national sodium intakes in 1990 and 2010: a systematic analysis of 24 h urinary sodium excretion and dietary surveys worldwide. BMJ Open. 2013 Dec 23;3(12):e003733.Thout SR, Santos JA, McKenzie B, Trieu K, Johnson C, McLean R, et al., The Science of Salt: Updating the evidence on global estimates of salt intake. J Clin Hypertens (Greenwich). 2019 Jun;21(6):710–21.Oyebode O, Oti S, Chen Y-F, Lilford RJ. Salt intakes in sub-Saharan Africa: a systematic review and meta-regression. Popul Health Metr. 2016;14:1NCD Risk Factor Collaboration (NCD-RisC). Worldwide trends in hypertension prevalence and progress in treatment and control from 1990 to 2019: a pooled analysis of 1201 population-representative studies with 104 million participants. Lancet. 2021 Sep 11;398(10304):957-980.Carrillo-Larco RM, Bernabe-Ortiz A. Sodium and Salt Consumption in Latin America and the Caribbean: A Systematic-Review and Meta-Analysis of Population-Based Studies and Surveys. Nutrients. 2020 Feb 20;12(2)Huang L, Crino M, Wu JHY, Woodward M, Barzi F, Land M-A, et al. Mean population salt intake estimated from 24-h urine samples and spot urine samples: a systematic review and meta-analysis. Int J Epidemiol. 2016 Feb;45(1):239–50.Santos JA, Li KC, Huang L, Mclean R, Petersen K, Di Tanna GL, et al. Change in mean salt intake over time using 24-h urine versus overnight and spot urine samples: a systematic review and meta-analysis. Nutr J. 2020 Dec 6;19(1):136.